# ReHLine: Regularized Composite ReLU-ReHU Loss Minimization with Linear Computation and Linear Convergence

**Ben Dai**[*]
Department of Statistics
The Chinese University of Hong Kong
bendai@cuhk.edu.hk

**Yixuan Qiu**[*][†]
School of Statistics and Management
Shanghai University of Finance and Economics
qiuyixuan@sufe.edu.cn

## Abstract

Empirical risk minimization (ERM) is a crucial framework that offers a general approach to handling a broad range of machine learning tasks. In this paper, we propose a novel algorithm, called ReHLine, for minimizing a set of regularized ERMs with convex piecewise linear-quadratic loss functions and optional linear constraints. The proposed algorithm can effectively handle diverse combinations of loss functions, regularization, and constraints, making it particularly well-suited for complex domain-specific problems. Examples of such problems include FairSVM, elastic net regularized quantile regression, Huber minimization, etc. In addition, ReHLine enjoys a provable *linear* convergence rate and exhibits a per-iteration computational complexity that scales *linearly* with the sample size. The algorithm is implemented with both Python and R interfaces, and its performance is benchmarked on various tasks and datasets. Our experimental results demonstrate that ReHLine significantly surpasses generic optimization solvers in terms of computational efficiency on large-scale datasets. Moreover, it also outperforms specialized solvers such as LIBLINEAR in SVMs, HQREG in Huber minimization and LIGHTNING (SAGA, SAG, SDCA, SVRG) in smoothed SVMs, exhibiting exceptional flexibility and efficiency. The source code, project page, accompanying software, and the Python/R interface can be accessed through the link: https://github.com/softmin/ReHLine.

## 1 Introduction

Empirical risk minimization (ERM) with different losses is a fundamental framework in statistical machine learning [43], which has demonstrated a significant impact on many application areas, such as artificial intelligence, computational biology, computer vision, natural language processing, electronic engineering, and finance. In fact, many common practical ERMs are associated with a piecewise linear-quadratic (PLQ; [34]) loss function, including support vector machines (SVMs; [7]), the least absolute deviation regression, quantile regression [26], Huber regression [22], borrowing cost function in portfolio selection [28], and pricing/planning problems in electricity and operational research [16, 1], among many others. In addition, ERMs are frequently blended with linear constraints, for instance, the fairness constraints in fair classification [48], monotonicity constraints in classification or regression, etc. More linear constraints according to the prior domain knowledge, such as the sum-to-one constraints in the portfolio variance minimization problem [24], negative or positive constraints in non-performing loan [36], are considered in various real applications. Therefore, it has become of utmost importance to develop fast and scalable numerical algorithms to solve ERMs with PLQ losses based on large-scale datasets and flexible linear constraints.

---

[*]Co-first authorship, and the order is determined by coin toss. [†]Corresponding author.

37th Conference on Neural Information Processing Systems (NeurIPS 2023).

In this paper, we consider a general regularized ERM based on a convex PLQ loss with linear constraints:

$$\min_{\boldsymbol{\beta} \in \mathbb{R}^d} \sum_{i=1}^{n} L_i(\mathbf{x}_i^{\intercal} \boldsymbol{\beta}) + \frac{1}{2} \|\boldsymbol{\beta}\|_2^2, \quad \text{s.t. } \mathbf{A}\boldsymbol{\beta} + \mathbf{b} \geq \mathbf{0}, \tag{1}$$

where $\mathbf{x}_i \in \mathbb{R}^d$ is the feature vector for the $i$-th observation, $\boldsymbol{\beta} \in \mathbb{R}^d$ is an unknown coefficient vector, $\mathbf{A} \in \mathbb{R}^{K \times d}$ and $\mathbf{b} \in \mathbb{R}^K$ are defined as linear inequality constraints for $\boldsymbol{\beta}$, and $L_i(\cdot) \geq 0$ is a convex PLQ loss function. The PLQ function class greatly generalizes existing popular loss functions, and the convexity of $L_i(\cdot)$ guarantees the global convergence of optimization. Here, we focus on working with a large-scale dataset, where the dimension of the coefficient vector and the total number of constraints are comparatively much smaller than the sample size, that is, $d \ll n$ and $K \ll n$.

Although (1) is a strongly convex problem with affine constraints, thus admitting a unique global optimum, directly solving (1) by a generic solver may encounter various challenges. For example, $L_i(\cdot)$ is in general non-smooth, so gradient-based methods may fail, and hence slow-convergent subgradient methods need to be used. In addition, the constraint set is a polyhedron, whose projection operator is non-trivial to compute, making various projection-based methods difficult to apply.

Of course, for some specific forms of $L_i(\cdot)$, highly efficient solvers for (1) indeed exist. One remarkable example is the LIBLINEAR solver [12], which has gained great success in solving large-scale SVMs. However, LIBLINEAR highly relies on the hinge loss function, so its success does not directly transfer to more general loss functions.

**Outline.** To this end, in this article we develop a novel algorithm to solve (1) with the greatest generality. The proposed algorithm, named ReHLine, has three key ingredients. First, we show that any convex PLQ loss function can be decomposed as the sum of a finite number of rectified linear units (ReLU, [15]) and rectified Huber units (ReHU, defined in Section 2). Second, based on this decomposition, the dual problem of (1) is shown to be a box-constrained quadratic programming (box-QP) problem. Finally, a special paired primal-dual coordinate descent algorithm is developed to simultaneously solve the primal and dual problems, with a computational cost linear in $n$.

We emphasize that for the second point, the dual box-QP problem is highly structured. Therefore, a general-purpose solver may not fully exploit the structure, thus leading to slower convergence rates or higher computational costs. As a direct comparison between the proposed ReHLine solver and existing algorithms, including the projected (sub-)gradient descent (P-GD; [6]), the coordinate descent (CD; [45]), the interior-point methods (IPMs; [13, 14, 18]), the stochastic dual coordinate ascent (SDCA; [39, 40]) related methods, and the alternating direction method of multipliers (ADMM; [5]), Table 1 summarizes the required number of iterations and per-iteration cost of each method. More details and discussions for this comparison are given in Section 4.

Table 1: Overview of existing algorithms for solving (1). Column COMPLEXITY (PER ITERATION) shows the computational complexity of the algorithm per iteration. Here, we focus only on the order of $n$ since $d \ll n$ is assumed in our setting. Column #ITERATIONS shows the number of iterations needed to achieve an accuracy of $\varepsilon$ to the minimizer.

| ALGORITHM | COMPLEXITY (PER ITERATION) | #ITERATIONS | COMPLEXITY (TOTAL) |
|---|---|---|---|
| P-GD | $\mathcal{O}(n)$ | $\mathcal{O}(\varepsilon^{-1})$ [6] | $\mathcal{O}(n\varepsilon^{-1})$ |
| CD | $\mathcal{O}(n^2)$ | $\mathcal{O}(\log(\varepsilon^{-1}))$ [31] | $\mathcal{O}(n^2 \log(\varepsilon^{-1}))$ |
| IPM | $\mathcal{O}(n^2)$ | $\mathcal{O}(\log(\varepsilon^{-1}))$ [18] | $\mathcal{O}(n^2 \log(\varepsilon^{-1}))$ |
| ADMM | $\mathcal{O}(n^2)$ | $o(\varepsilon^{-1})$ [9, 20] | $o(n^2\varepsilon^{-1})$ |
| SDCA | $\mathcal{O}(n)$ | $\mathcal{O}(\varepsilon^{-1})$ [39] | $\mathcal{O}(n\varepsilon^{-1})$ |
| ReHLine (ours) | $\mathcal{O}(n)$ | $\mathcal{O}(\log(\varepsilon^{-1}))$ | $\mathcal{O}(n \log(\varepsilon^{-1}))$ |

**Contribution.** Compared with other specialized ERM solvers or general-purpose box-QP solvers, the proposed ReHLine solver has four appealing "linear properties":

1. It applies to any convex piecewise linear-quadratic loss function (potential for non-smoothness included), including the hinge loss, the check loss, the Huber loss, etc.
2. In addition, it supports linear equality and inequality constraints on the parameter vector.
3. The optimization algorithm has a provable linear convergence rate.
4. The per-iteration computational complexity is linear in the sample size.

Moreover, ReHLine is designed to be a computationally efficient and practically useful software package for large-scale ERM problems. As an example, our experiments in Section 5 have shown that ReHLine significantly surpasses generic solvers in terms of computational efficiency on large-scale datasets, and provides reasonable improvements over specialized solvers and problems, such as LIBLINEAR, HQREG and LIGHTNING, given that LIBLINEAR, HQREG and LIGHTNING are one of the fastest and most widely-used solvers for SVM, Huber minimization, and smoothed SVM, respectively. Finally, ReHLine is available as user-friendly software in both Python and R, which facilitates ease of use for researchers, practitioners, and a wider audience.

## 2 The ReLU-ReHU decomposition

As is introduced in Section 1, we attempt to solve ERM with a general loss function, and one sensible choice is the class of convex PLQ functions. A univariate loss function $f : \mathbb{R} \to \mathbb{R}_{\geq 0}$ is PLQ if there exist a finite number of knots $t_1 < t_2 < \cdots < t_K$ such that $f$ is equal to a linear or quadratic function on each of the intervals $(-\infty, t_1], [t_1, t_2], \ldots, [t_{K-1}, t_K], [t_K, \infty)$. In what follows we merely consider nonnegative convex PLQ functions, which also imply the continuity of the loss function. It is well known that PLQ functions are universal approximators of continuous functions [38], but the canonical definition of PLQ is not convenient for optimization algorithms. Instead, in this section we prove that convex PLQ functions can actually be decomposed into a series of simple and basic units.

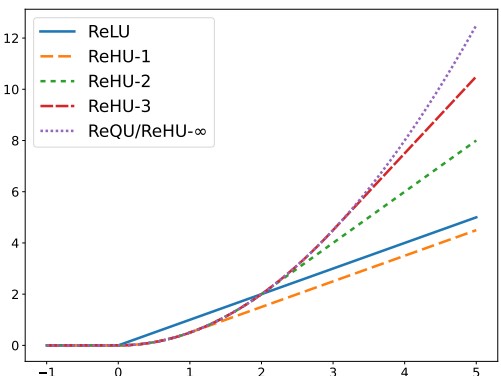

Figure 1: The ReLU function and the ReHU function with $\tau = 1, 2, 3, \infty$.

Before proceeding, we introduce a generalized quadratic version of ReLU, the so-called rectified Huber unit, ReHU:

$$\text{ReHU}_\tau(z) = \begin{cases} 0, & z \leq 0 \\ z^2/2, & 0 < z \leq \tau \\ \tau(z - \tau/2), & z > \tau \end{cases} \tag{2}$$

where $\tau \in [0, \infty]$ is a hyperparameter of ReHU switching from quadratic to linear form, see examples in Figure 1. ReHU also incorporates the rectified quadratic unit (ReQU) as a special case, that is, $\text{ReHU}_\infty(z) = \text{ReQU}(z)/2 = (z_+)^2/2$ when $\tau = \infty$, where $z_+ = \max\{z, 0\}$. Then we give the characterization of a composite ReLU-ReHU function.

**Definition 1** (Composite ReLU-ReHU function). *A function $L(z)$ is composite ReLU-ReHU, if there exist $\mathbf{u}, \mathbf{v} \in \mathbb{R}^L$ and $\boldsymbol{\tau}, \mathbf{s}, \mathbf{t} \in \mathbb{R}^H$ such that*

$$L(z) = \sum_{l=1}^{L} \text{ReLU}(u_l z + v_l) + \sum_{h=1}^{H} \text{ReHU}_{\tau_h}(s_h z + t_h), \tag{3}$$

*where $\text{ReLU}(z) = z_+$, and $\text{ReHU}_{\tau_h}(z)$ is defined in (2).*

Clearly, the composite ReLU-ReHU function class is closed under affine transformations, as is indicated by Proposition 1.

**Proposition 1** (Closure under affine transformation). *If $L(z)$ is a composite ReLU-ReHU function as in (3), then for any $c > 0$, $p \in \mathbb{R}$, and $q \in \mathbb{R}$, $cL(pz + q)$ is also composite ReLU-ReHU, that is,*

$$cL(pz + q) = \sum_{l=1}^{L} \text{ReLU}(u'_l z + v'_l) + \sum_{h=1}^{H} \text{ReHU}_{\tau'_h}(s'_h z + t'_h),$$

*where $u'_l = cpu_l$, $v'_l = cu_l q + cv_l$, $\tau'_h = \sqrt{c}\tau_h$, $s'_h = \sqrt{c}ps_h$, and $t'_h = \sqrt{c}(s_h q + t_h)$.*

More importantly, we show that the composite ReLU-ReHU function class is equivalent to the class of convex PLQ functions.

**Theorem 1.** *A loss function* $L : \mathbb{R} \to \mathbb{R}_{\geq 0}$ *is convex PLQ if and only if it is composite ReLU-ReHU.*

Based on the fact that the loss functions of various SVMs (the hinge loss, the squared hinge loss, and the smoothed hinge loss), quantile regression (the check loss), and least absolute deviation (the absolute loss) are all convex PLQs, Theorem 1 suggests that Definition 1 incorporates numerous widely-used losses in statistics and machine learning applications. Table 2 provides a convenient translation from some commonly-used loss functions to their ReLU-ReHU representation.

Table 2: Some widely-used composite ReLU-ReHU losses as in (3). Here SVM is weighted SVMs based on the hinge loss [7], sSVM is smoothed SVMs based on the smoothed hinge loss [33], $\text{SVM}^2$ is weighted squared SVMs based on the squared hinge loss [7], LAD is the least absolute deviation regression, SVR is support vector regression with the $\varepsilon$-insensitive loss [44], and QR is quantile regression with the check loss [26].

| PROBLEM | LOSS ($L_i(z_i)$) | COMPOSITE ReLU-ReHU PARAMETERS |
|---|---|---|
| SVM | $c_i(1 - y_i z_i)_+$ | $u_{1i} = -c_i y_i,\ v_{1i} = c_i$ |
| sSVM | $c_i \text{ReHU}_1(-(y_i z_i - 1))$ | $s_{1i} = -\sqrt{c_i} y_i,\ t_{1i} = \sqrt{c_i},\ \tau = \sqrt{c_i}$ |
| $\text{SVM}^2$ | $c_i((1 - y_i z_i)_+)^2$ | $s_{1i} = -\sqrt{2c_i} y_i,\ t_{1i} = \sqrt{2c_i},\ \tau = \infty$ |
| LAD | $c_i \lvert y_i - z_i \rvert$ | $u_{1i} = c_i,\ v_{1i} = -c_i y_i,\ u_{2i} = -c_i,\ v_{2i} = c_i y_i$ |
| SVR | $c_i(\lvert y_i - z_i \rvert - \varepsilon)_+$ | $u_{1i} = c_i,\ v_{1i} = -(y_i + \varepsilon),\ u_{2i} = -c_i,\ v_{2i} = y_i - \varepsilon$ |
| QR | $c_i \rho_\kappa(y_i - z_i)$ (A.3) | $u_{1i} = -c_i \kappa,\ v_{1i} = \kappa c_i y_i,\ u_{2i} = c_i(1 - \kappa),\ v_{2i} = -c_i(1 - \kappa) y_i$ |

Table 3: Some widely-used linear constraints of the form $\mathbf{A}\boldsymbol{\beta} + \mathbf{b} \geq 0$, where U-bidiag($[-1, 1]$) is an upper bidiagonal matrix with main diagonal being -1 and upper diagonal being 1.

| CONSTRAINT | FORM | REHLINE PARAMETERS |
|---|---|---|
| Nonnegative | $\boldsymbol{\beta} \geq \mathbf{0}$ | $\mathbf{A} = -\mathbf{I},\ \mathbf{b} = \mathbf{0}$ |
| Box | $\mathbf{l} \leq \boldsymbol{\beta} \leq \mathbf{u}$ | $\mathbf{A} = [\mathbf{I}; -\mathbf{I}],\ \mathbf{b} = [-\mathbf{l}; \mathbf{u}]$ |
| Monotonicity | $\beta_1 \leq \cdots \leq \beta_d$ | $\mathbf{A} = \mathbf{B}_{1:d-1},\ \mathbf{B} = \text{U-bidiag}([-1, 1]),\ \mathbf{b} = \mathbf{0}_{d-1}$ |
| Fairness (A.1) | $\lvert \mathbf{Z}^\mathsf{T} \mathbf{X} \boldsymbol{\beta} / n \rvert \leq \boldsymbol{\rho}$ | $\mathbf{A} = \mathbf{Z}^\mathsf{T} \mathbf{X} [\mathbf{I}; -\mathbf{I}] / n,\ \mathbf{b} = [\boldsymbol{\rho}; \boldsymbol{\rho}]$ |

Taken together, our algorithm is designed to address the empirical regularized ReLU-ReHU minimization problem, named *ReHLine optimization*, of the following form:

$$\min_{\boldsymbol{\beta} \in \mathbb{R}^d} \sum_{i=1}^{n} \sum_{l=1}^{L} \text{ReLU}(u_{li} \mathbf{x}_i^\mathsf{T} \boldsymbol{\beta} + v_{li}) + \sum_{i=1}^{n} \sum_{h=1}^{H} \text{ReHU}_{\tau_{hi}}(s_{hi} \mathbf{x}_i^\mathsf{T} \boldsymbol{\beta} + t_{hi}) + \frac{1}{2}\|\boldsymbol{\beta}\|_2^2,$$
$$\text{s.t. } \mathbf{A}\boldsymbol{\beta} + \mathbf{b} \geq \mathbf{0}, \tag{4}$$

where $\mathbf{U} = (u_{li}), \mathbf{V} = (v_{li}) \in \mathbb{R}^{L \times n}$ and $\mathbf{S} = (s_{hi}), \mathbf{T} = (t_{hi}), \boldsymbol{\tau} = (\tau_{hi}) \in \mathbb{R}^{H \times n}$ are the ReLU-ReHU loss parameters, as illustrated in Table 2, and $(\mathbf{A}, \mathbf{b})$ are the constraint parameters, as illustrated in Table 3. This formulation has a wide range of applications, including statistics, machine learning, computational biology, and social studies. Some popular examples include SVMs with fairness constraints (FairSVM), elastic net regularized quantile regression (ElasticQR), ridge Huber minimization (RidgeHuber), and smoothed SVMs (sSVM). See Appendix A for details.

Despite appearing to impose limitations on the $l_2$-regularization in (4), our ReHLine formulation can actually be extended to more general regularization forms, including the elastic net penalty [49].

**Proposition 2.** *If* $L_i$ *is a composite ReLU-ReHU loss function with parameters* $(u_{li}, v_{li})_{l=1}^{L}$ *and* $(s_{hi}, t_{hi}, \tau_{hi})_{h=1}^{H}$, *then its elastic net regularized minimization with* $\lambda_1 \geq 0$ *and* $\lambda_2 > 0$,

$$\min_{\boldsymbol{\beta} \in \mathbb{R}^d} \sum_{i=1}^{n} L_i(\mathbf{x}_i^\mathsf{T} \boldsymbol{\beta}) + \lambda_1 \|\boldsymbol{\beta}\|_1 + \frac{\lambda_2}{2}\|\boldsymbol{\beta}\|_2^2, \quad \text{s.t.} \quad \mathbf{A}\boldsymbol{\beta} + \mathbf{b} \geq \mathbf{0},$$

*can be rewritten as the form of ReHLine optimization* (4) *with*

$$\mathbf{U} \leftarrow \begin{pmatrix} \frac{1}{\lambda_2}\mathbf{U} & \mathbf{0}_{L\times d} \\ \mathbf{0}_n^\mathsf{T} & \frac{\lambda_1}{\lambda_2}\mathbf{1}_d^\mathsf{T} \\ \mathbf{0}_n^\mathsf{T} & -\frac{\lambda_1}{\lambda_2}\mathbf{1}_d^\mathsf{T} \end{pmatrix}, \quad \mathbf{V} \leftarrow \begin{pmatrix} \frac{1}{\lambda_2}\mathbf{V} & \mathbf{0}_{L\times d} \\ \mathbf{0}_{n+d}^\mathsf{T} \\ \mathbf{0}_{n+d}^\mathsf{T} \end{pmatrix}, \quad \mathbf{X} \leftarrow \begin{pmatrix} \mathbf{X} \\ \mathbf{I}_d \end{pmatrix},$$

$$\mathbf{S} \leftarrow \begin{pmatrix} \sqrt{\frac{1}{\lambda_2}}\mathbf{S} & \mathbf{0}_d^\mathsf{T} \end{pmatrix}, \qquad \mathbf{T} \leftarrow \begin{pmatrix} \sqrt{\frac{1}{\lambda_2}}\mathbf{T} & \mathbf{0}_d^\mathsf{T} \end{pmatrix}, \qquad \boldsymbol{\tau} \leftarrow \begin{pmatrix} \sqrt{\frac{1}{\lambda_2}}\boldsymbol{\tau} & \mathbf{0}_d^\mathsf{T} \end{pmatrix},$$

*where $\leftarrow$ indicates assigning a value to a parameter.*

## 3 Optimization algorithm

In this section, we present the ReHLine algorithm to tackle the optimization in (4). The proposed algorithm is inspired by LIBLINEAR [12], which has gained great success in solving standard SVMs. The key idea of ReHLine is to leverage the linear property of Karush–Kuhn–Tucker (KKT) conditions, thus simplifying the iterative complexity of CD and achieving rapid convergence.

### 3.1 Primal and dual formulations

To proceed, we first demonstrate equivalent formulations for ReLU and ReHU, respectively, where the equivalence means that the value on the left is equal to the minimum value of the right-hand side:

$$\text{ReLU}(z) \iff \begin{array}{ll} \min\limits_{\pi} & \pi \\ \text{s.t.} & \pi \geq z, \\ & \pi \geq 0. \end{array} \qquad \text{ReHU}(z) \iff \begin{array}{ll} \min\limits_{\theta,\sigma} & \frac{1}{2}\theta^2 + \tau\sigma \\ \text{s.t.} & \theta + \sigma \geq z, \\ & \sigma \geq 0. \end{array} \tag{5}$$

It is worth noting that when $\tau = \infty$, the optimal solution for $\sigma$ is 0, and hence the last constraint ($\sigma \geq 0$) can be removed. Then according to the definition of the composite ReLU-ReHU loss function and the equivalent forms in (5), the ReHLine optimization problem (4) can be rewritten as:

$$\min_{\boldsymbol{\beta},\boldsymbol{\Pi},\boldsymbol{\Theta},\boldsymbol{\Sigma}} \sum_{i=1}^{n}\sum_{l=1}^{L}\pi_{li} + \sum_{i=1}^{n}\sum_{h=1}^{H}\frac{1}{2}\theta_{hi}^2 + \sum_{i=1}^{n}\sum_{h=1}^{H}\tau_{hi}\sigma_{hi} + \frac{1}{2}\|\boldsymbol{\beta}\|_2^2$$

$$\text{s.t.} \quad \mathbf{A}\boldsymbol{\beta} + \mathbf{b} \geq \mathbf{0}, \quad \pi_{li} \geq u_{li}\mathbf{x}_i^\mathsf{T}\boldsymbol{\beta} + v_{li}, \quad \theta_{hi} + \sigma_{hi} \geq s_{hi}\mathbf{x}_i^\mathsf{T}\boldsymbol{\beta} + t_{hi},$$

$$\pi_{li} \geq 0, \quad \sigma_{hi} \geq 0, \quad \text{for all } (i,l,h), \tag{6}$$

where $\boldsymbol{\Pi} = (\pi_{li}) \in \mathbb{R}^{L\times n}$, $\boldsymbol{\Theta} = (\theta_{hi}) \in \mathbb{R}^{H\times n}$ and $\boldsymbol{\Sigma} = (\sigma_{hi}) \in \mathbb{R}^{H\times n}$ are slack variables. Theorem 2 shows the dual form of the primal formulation (6) by using the Lagrangian duality theory, and the relation between primal and dual variables is also provided.

**Theorem 2.** *The Lagrangian dual problem of* (6) *is:*

$$(\widehat{\boldsymbol{\xi}}, \widehat{\boldsymbol{\Lambda}}, \widehat{\boldsymbol{\Gamma}}) = \underset{\boldsymbol{\xi},\boldsymbol{\Lambda},\boldsymbol{\Gamma}}{\text{argmin}} \quad \mathcal{L}(\boldsymbol{\xi},\boldsymbol{\Lambda},\boldsymbol{\Gamma})$$

$$\text{s.t.} \quad \boldsymbol{\xi} \geq \mathbf{0}, \quad \mathbf{E} \geq \boldsymbol{\Lambda} \geq \mathbf{0}, \quad \boldsymbol{\tau} \geq \boldsymbol{\Gamma} \geq \mathbf{0}, \tag{7}$$

$$\mathcal{L}(\boldsymbol{\xi},\boldsymbol{\Lambda},\boldsymbol{\Gamma}) = \frac{1}{2}\boldsymbol{\xi}^\mathsf{T}\mathbf{A}\mathbf{A}^\mathsf{T}\boldsymbol{\xi} + \frac{1}{2}vec(\boldsymbol{\Lambda})^\mathsf{T}\bar{\mathbf{U}}_{(3)}^\mathsf{T}\bar{\mathbf{U}}_{(3)}vec(\boldsymbol{\Lambda}) + \frac{1}{2}vec(\boldsymbol{\Gamma})^\mathsf{T}\big(\bar{\mathbf{S}}_{(3)}^\mathsf{T}\bar{\mathbf{S}}_{(3)} + \mathbf{I}\big)vec(\boldsymbol{\Gamma})$$

$$- \boldsymbol{\xi}^\mathsf{T}\mathbf{A}\bar{\mathbf{U}}_{(3)}vec(\boldsymbol{\Lambda}) - \boldsymbol{\xi}^\mathsf{T}\mathbf{A}\bar{\mathbf{S}}_{(3)}vec(\boldsymbol{\Gamma}) + vec(\boldsymbol{\Lambda})^\mathsf{T}\bar{\mathbf{U}}_{(3)}^\mathsf{T}\bar{\mathbf{S}}_{(3)}vec(\boldsymbol{\Gamma})$$

$$+ \boldsymbol{\xi}^\mathsf{T}\mathbf{b} - \text{Tr}(\boldsymbol{\Lambda}\mathbf{V}^\mathsf{T}) - \text{Tr}(\boldsymbol{\Gamma}\mathbf{T}^\mathsf{T}), \tag{8}$$

*where $\boldsymbol{\xi} \in \mathbb{R}^K$, $\boldsymbol{\Lambda} = (\lambda_{li}) \in \mathbb{R}^{L\times n}$, and $\boldsymbol{\Gamma} = (\gamma_{hi}) \in \mathbb{R}^{H\times n}$ are dual variables, $\bar{\mathbf{U}}_{(3)} \in \mathbb{R}^{d\times nL}$ and $\bar{\mathbf{S}}_{(3)} \in \mathbb{R}^{d\times nH}$ are the mode-3 unfolding of the tensors $\bar{\mathbf{U}} = (u_{lij}) \in \mathbb{R}^{L\times n\times d}$ and $\bar{\mathbf{S}} = (s_{hij}) \in \mathbb{R}^{H\times n\times d}$, respectively, $u_{lij} = u_{li}x_{ij}$, $s_{hij} = s_{hi}x_{ij}$, $\mathbf{I}$ is the identity matrix, and all inequalities are elementwise.*

*Moreover, the optimal point $\widehat{\boldsymbol{\beta}}$ of* (6) *can be recovered as:*

$$\widehat{\boldsymbol{\beta}} = \sum_{k=1}^{K}\widehat{\xi}_k\mathbf{a}_k - \sum_{i=1}^{n}\mathbf{x}_i\left(\sum_{l=1}^{L}\widehat{\lambda}_{li}u_{li} + \sum_{h=1}^{H}\widehat{\gamma}_{hi}s_{hi}\right) = \mathbf{A}^\mathsf{T}\widehat{\boldsymbol{\xi}} - \bar{\mathbf{U}}_{(3)}vec(\widehat{\boldsymbol{\Lambda}}) - \bar{\mathbf{S}}_{(3)}vec(\widehat{\boldsymbol{\Gamma}}). \tag{9}$$

## 3.2 ReHLine update rules

Indeed, (9) is a KKT condition of (6), and the main idea of the proposed ReHLine solver is to update the dual variables $(\boldsymbol{\xi}, \boldsymbol{\Lambda}, \boldsymbol{\Gamma})$ by CD on (7) with an *analytic solution*, and *simultaneously* update the primal variable $\boldsymbol{\beta}$ by the KKT condition (9). Most importantly, the computational complexity of the CD updates on (7) can be significantly reduced by using the information of $\boldsymbol{\beta}$.

**Canonical CD updates.** As a first step, we consider the canonical CD update rule that directly optimizes the dual problem (7) with respect to a single variable. For brevity, in this section we only illustrate the result for $\lambda_{li}$ variables, and the full details are given in Appendix B.

By excluding the terms unrelated to $\lambda_{li}$, we have $\lambda_{li}^{\text{new}} = \text{argmin}_{0 \le \lambda \le 1} \mathcal{L}_{li}(\lambda)$, where

$$\mathcal{L}_{li}(\lambda) = \frac{1}{2} u_{li}^2 (\mathbf{x}_i^\mathsf{T} \mathbf{x}_i) \lambda^2 + \sum_{(l',i') \neq (l,i)} \lambda_{l'i'} u_{l'i'} u_{li} (\mathbf{x}_{i'}^\mathsf{T} \mathbf{x}_i) \lambda - \sum_{k=1}^{K} \xi_k u_{li} (\mathbf{a}_k^\mathsf{T} \mathbf{x}_i) \lambda$$
$$+ \sum_{h',i'} u_{li} \gamma_{h'i'} s_{h'i'} \mathbf{x}_i^\mathsf{T} \mathbf{x}_{i'} \lambda - v_{li} \lambda.$$

Therefore, by simple calculations we obtain

$$\lambda_{li}^{\text{new}} = \mathcal{P}_{[0,1]} \left( \frac{u_{li} \mathbf{x}_i^\mathsf{T} \left( \sum_{k=1}^{K} \xi_k \mathbf{a}_k - \sum_{(l',i') \neq (l,i)} \lambda_{l'i'} u_{l'i'} \mathbf{x}_{i'} - \sum_{h',i'} \gamma_{h'i'} s_{h'i'} \mathbf{x}_{i'} \right) + v_{li}}{u_{li}^2 \|\mathbf{x}_i\|_2^2} \right),$$
(10)

where $\mathcal{P}_{[a,b]}(x) = \max(a, \min(b, x))$ means projecting a real number $x$ to the interval $[a, b]$.

Clearly, assuming the values $\mathbf{x}_i^\mathsf{T} \mathbf{a}_k$ and $\|\mathbf{x}_i\|_2^2$ are cached, updating one $\lambda_{li}$ value requires $\mathcal{O}(K + nd + nL + nH)$ of computation, and updating the whole $\boldsymbol{\Lambda}$ matrix requires $\mathcal{O}(nL(K + nd + nL + nH))$. Adding all variables together, the canonical CD update rule for one full cycle has a computational complexity of $\mathcal{O}((K + nd + nL + nH)(K + nL + nH))$.

**ReHLine updates.** The proposed ReHLine algorithm, on the other hand, significantly reduces the computational complexity of canonical CD by updating $\boldsymbol{\beta}$ according to the KKT condition (9) after each update of a dual variable. To see this, let $\boldsymbol{\mu} := (\boldsymbol{\xi}, \boldsymbol{\Lambda}, \boldsymbol{\Gamma})$ denote all the dual variables, and define

$$\boldsymbol{\beta}(\boldsymbol{\mu}) = \sum_{k=1}^{K} \xi_k \mathbf{a}_k - \sum_{i=1}^{n} \mathbf{x}_i \left( \sum_{l=1}^{L} \lambda_{li} u_{li} + \sum_{h=1}^{H} \gamma_{hi} s_{hi} \right).$$

Then it can be proved that $(\nabla_\lambda \mathcal{L}_{li})(\lambda_{li}) = -(u_{li} \mathbf{x}_i^\mathsf{T} \boldsymbol{\beta}(\boldsymbol{\mu}) + v_{li})$. Therefore, when $\boldsymbol{\mu}$ is fixed at $\boldsymbol{\mu}^{\text{old}} = (\boldsymbol{\xi}^{\text{old}}, \boldsymbol{\Lambda}^{\text{old}}, \boldsymbol{\Gamma}^{\text{old}})$ and let $\boldsymbol{\beta}^{\text{old}} = \boldsymbol{\beta}(\boldsymbol{\mu}^{\text{old}})$, (10) can be rewritten as

$$\lambda_{li}^{\text{new}} = \mathcal{P}_{[0,1]} \left( \lambda_{li}^{\text{old}} - \frac{(\nabla_{\lambda_{li}} \mathcal{L})(\lambda^{\text{old}})}{u_{li}^2 \|\mathbf{x}_i\|_2^2} \right) = \mathcal{P}_{[0,1]} \left( \lambda_{li}^{\text{old}} + \frac{u_{li} \mathbf{x}_i^\mathsf{T} \boldsymbol{\beta}^{\text{old}} + v_{li}}{u_{li}^2 \|\mathbf{x}_i\|_2^2} \right).$$

Accordingly, the primal variable $\boldsymbol{\beta}$ is updated as

$$\boldsymbol{\beta}^{\text{new}} = \boldsymbol{\beta}^{\text{old}} - (\lambda_{li}^{\text{new}} - \lambda_{li}^{\text{old}}) u_{li} \mathbf{x}_i,$$

which can then be used for the next dual variable update. Simple calculations show that this scheme only costs $\mathcal{O}(d)$ of computation for one $\lambda_{li}$ variable.

The update rules for other dual variables are similar, with the complete algorithm shown in Algorithm 1 and the full derivation details given in Appendix B. Overall, the ReHLine update rule has a computational complexity of $\mathcal{O}((K + nL + nH)d)$ for one full cycle. Given that $(K, L, H)$ are all small numbers, the complexity is *linear* with respect to both the sample size $n$ and the dimension $d$.

We emphasize that the linear relationship (9) between the primal and dual variables, as well as its consequence of reducing the computational cost, is highly non-trivial. This technique has been proposed for specialized problems such as SVMs via the LIBLINEAR solver [21], and here we show that its success can be greatly generalized to any convex PLQ loss functions.

**Algorithm 1:** The ReHLine algorithm that solves (4).

**Input** : $\mathbf{X} \in \mathbb{R}^{n \times d}$, $\mathbf{U}, \mathbf{V} \in \mathbb{R}^{L \times n}$, $\mathbf{S}, \mathbf{T}, \boldsymbol{\tau} \in \mathbb{R}^{H \times n}$, $\mathbf{A} \in \mathbb{R}^{K \times d}$, $\mathbf{b} \in \mathbb{R}^K$

1 Initialize $\boldsymbol{\xi} \geq \mathbf{0}$, $\mathbf{E} \geq \boldsymbol{\Lambda} \geq \mathbf{0}$, $\boldsymbol{\tau} \geq \boldsymbol{\Gamma} \geq \mathbf{0}$;

2 Compute $\boldsymbol{\beta} = \mathbf{A}^{\mathsf{T}} \boldsymbol{\xi} - \sum_{i=1}^{n} \mathbf{x}_i \left( \sum_{l=1}^{L} \lambda_{li} u_{li} + \sum_h^H \gamma_{hi} s_{hi} \right)$;

3 Compute and store $\mathbf{r} = (r_i) \in \mathbb{R}^n$, $r_i = \|\mathbf{x}_i\|_2^2$, and $\mathbf{p} = (p_k) \in \mathbb{R}^K$, $p_k = \|\mathbf{a}_k\|_2^2$;

4 **while** *maximum iteration* **do**

5      **for** $k \in \mathcal{K} = [K]$ **do**

6          $\varepsilon^* \leftarrow \max\left(-\xi_k, -p_k^{-1}(\mathbf{a}_k^{\mathsf{T}} \boldsymbol{\beta} + b_k)\right);$    $\xi_k \leftarrow \xi_k + \varepsilon^*;$    $\boldsymbol{\beta} \leftarrow \boldsymbol{\beta} + \varepsilon^* \mathbf{a}_k;$

7      **for** $(l, i) \in \mathcal{L} = [L] \bigotimes [n]$ **do**

8          $\varepsilon^* \leftarrow \max\left(-\lambda_{li}, \min\left(1 - \lambda_{li}, u_{li}^{-2} r_i^{-1}(v_{li} + u_{li} \mathbf{x}_i^{\mathsf{T}} \boldsymbol{\beta})\right)\right);$
         $\lambda_{li} \leftarrow \lambda_{li} + \varepsilon^*;$    $\boldsymbol{\beta} \leftarrow \boldsymbol{\beta} - \varepsilon^* u_{li} \mathbf{x}_i;$

9      **for** $(h, i) \in \mathcal{H} = [H] \bigotimes [n]$ **do**

10         $\varepsilon^* \leftarrow \max\left(-\gamma_{hi}, \min\left(\tau_{hi} - \gamma_{hi}, (s_{hi}^2 r_i + 1)^{-1}(t_{hi} + s_{hi} \mathbf{x}_i^{\mathsf{T}} \boldsymbol{\beta} - \gamma_{hi})\right)\right);$
         $\gamma_{hi} \leftarrow \gamma_{hi} + \varepsilon^*;$    $\boldsymbol{\beta} \leftarrow \boldsymbol{\beta} - \varepsilon^* s_{hi} \mathbf{x}_i;$

**Output** : $\boldsymbol{\xi}$, $\boldsymbol{\Lambda}$, $\boldsymbol{\Gamma}$, $\boldsymbol{\beta}$

### 3.3 Global convergence rate

Finally, we show that the ReHLine algorithm has a fast global linear convergence rate.

**Theorem 3.** *Let $\boldsymbol{\mu}^{(q)} := (\boldsymbol{\xi}^{(q)}, \boldsymbol{\Lambda}^{(q)}, \boldsymbol{\Gamma}^{(q)})$ be a sequence of iterates generated by Algorithm 1, and $\boldsymbol{\mu}^*$ be a minimizer of the dual objective* (8). *Then $\boldsymbol{\mu}^{(q)}$ is feasible, and the dual objective value converges at least linearly to that of $\boldsymbol{\mu}^*$, that is, there exist $0 < \eta < 1$ and $q_0$ such that for all $q \geq q_0$,*

$$\mathcal{L}(\boldsymbol{\mu}^{(q+1)}) - \mathcal{L}(\boldsymbol{\mu}^*) \leq \eta\left(\mathcal{L}(\boldsymbol{\mu}^{(q)}) - \mathcal{L}(\boldsymbol{\mu}^*)\right).$$

## 4 Related work

Table 1 summarizes the existing methods in solving the primal or dual problem of (1). Note that Column #ITERATIONS is the best-fit theoretical results of convergence analysis for each algorithm based on our literature search, which may be improved with further developments of convergence analysis. These algorithms can be broadly divided into the following categories.

**Generic QP solvers.** For instance, a projected sub-gradient descent is readily applicable to the primal form (1) by computing the sub-gradient of each loss and the projection under affine constraints, although the projection operator itself is already difficult. As for the box-QP dual problem (7), CD converges linearly to the optimal solution [27, 31, 35, 42]. Box-QP can also be efficiently solved using IPM, which leverages second-order cone programming techniques [13, 14]. Moreover, ADMM was employed to iteratively minimize the augmented Lagrangian of (1).

**ERM solvers.** There are multiple existing works developed to tackle ERM problems, such as SAG [37], SVRG [25], SAGA [8], proximal SDCA [40] and SDCA [39]. However, SAG, SVRG, SAGA and proximal SDCA can only handle smooth loss functions, with an optional non-smooth term that has an easy proximal operator. SDCA applies to general loss functions, but it requires the convex conjugate of loss functions, which is not necessarily simple to compute, also see discussion in Section 3.1 in [40]. Moreover, it only guarantees a sublinear convergence rate for non-smooth loss functions. In contrast, ReHLine supports all convex PLQ loss functions with optional general linear constraints, which could be non-smooth by construction, and they all enjoy a linear computational complexity and provable linear convergence.

For non-smooth loss functions, another existing method to solve ERM is the smoothing technique [40, 3, 30], which approximates the non-smooth terms by smooth functions, and then uses smooth-based algorithms to solve the smooth problem. However, the choice of the smoothing function and smoothing parameter typically requires additional knowledge, and the linear convergence rate is not always guaranteed.

Despite the successful use of the existing algorithms in solving (1), these methods either suffer from slow convergence (P-GD), or high computational costs (CD, IPM, and ADMM). This is partly because these general methods do not fully exploit the inherent structure of the problem. In contrast, ReHLine leverages the KKT condition in (9) to substantially improve numerical performance, thus achieving a linear computational complexity per iteration with respect to the sample size, as well as a linear convergence rate.

In addition to generic algorithms, specialized algorithms have been developed for specific types of (1). For example, the dual coordinate descent of LIBLINEAR in solving SVMs [21], the semismooth Newton coordinate descent algorithm of HQREG in solving the ridge regularized Huber minimization [46], and SAGA, SAG, proximal SDCA, and SVRG of LIGHTNING in solving smooth ERMs [4], such as smoothed SVMs. Our algorithm is also compared to these specialized methods in Section 5.

**Software.** Below are some commonly-used software for solving (1) based on the algorithms available.

- CVX/CVXPY [17, 10]: a modeling language for convex optimization problems and supports a wide range of solvers such as ECOS, SCS, OSQP, and MOSEK.
- MOSEK [2]: a commercial convex optimization solver based on IPM.
- ECOS [11]: an open-source solver of second-order cone programming based on IPM.
- SCS [32]: an open-source convex quadratic cone solver based on ADMM.
- DCCP [41]: a disciplined convex-concave programming solver built on top of CVXPY, which has been used to solve classification problems with fairness constraints [47].
- LIBLINEAR [21]: a library for large linear classification.
- HQREG [46]: an R package solving Lasso or elastic net penalized Huber loss regression and quantile regression.
- LIGHTNING [4]: a Python library solving various ERMs based on primal SDCA, SGD, AdaGrad, SAG, SAGA, SVRG.

## 5 Experiments and benchmarks

In this section we demonstrate the performance of ReHLine compared with the existing state-of-the-art (SOTA) solvers on various machine learning tasks. The source code of ReHLine, along with Python/R interface, is readily accessible on our GitHub repository (https://github.com/softmin/ReHLine) and our project page (https://rehline.github.io/). Our experiments involve five prominent machine learning tasks that draw from diverse sets of data. Specifically, we focus on four classification datasets and five regression datasets sourced from OpenML (https://www.openml.org/), with various scales and dimensions. The experiment settings are summarized in Table 4. To achieve a fair comparison, we use a well-organized toolset and framework, the BENCHOPT library [29], to implement optimization benchmarks for all the SOTA solvers.

Due to representation limits, we only present the results using a variant of the ReHLine algorithm that supports variable shrinkage (refer to Algorithm 2). Results utilizing ReHLine with or without shrinkage are reported on our benchmark repository (https://github.com/softmin/ReHLine-benchmark). Note that there is no substantial difference between these variations, and they do not influence the conclusions drawn from our experiments. Simultaneously, in the benchmark repository, we provide a more detailed "time versus objective" plot for each task/dataset and each solver produced by BENCHOPT.

Table 4: The classification and regression datasets with different scales used in our experiments.

| CLASSIFICATION DATASET ($n \times d$) | REGRESSION DATASET ($n \times d$) |
| --- | --- |
| Steel-plates-fault (SPF): $1941 \times 34$ | Liver-disorders (LD): $345 \times 6$ |
| Philippine: $5832 \times 309$ | Kin8nm: : $8191 \times 9$ |
| Sylva-prior: $14395 \times 109$ | Topo-2-1: $8885 \times 267$ |
| Creditcard: $284807 \times 31$ | House-8L: $22784 \times 9$ |
| | Buzzin-Twitter (BT): $583250 \times 78$ |

For the **FairSVM** task, ReHLine is compared with the original solver [48] based on DCCP and other generic solvers in CVXPY. In this case, both feasibility and optimality are examined, where the

optimality is measured by the objective function, and the feasibility is measured by the violation of constraints: $\max \left(n^{-1}\left|\sum_{i=1}^{n} z_i \boldsymbol{\beta}^{\mathsf{T}} \mathbf{x}_i\right| - \rho, 10^{-6}\right)$. Moreover, we examine the performance of **ElasticQR** by considering the model defined in (A.2) with $\lambda_1 = \lambda_2 = 1$, **RidgeHuber** of (A.4) with $\lambda_1 = 0, \lambda_2 = 1$, and **SVM** of (A.1) and **sSVM** of (A.5) with $C = 1$. Table 5 presents the running times in seconds of all solvers that converge based on a relative/absolute tolerance of $10^{-5}$.

Table 5: The averaged running times ($\pm$ standard deviation) of SOTA solvers on machine learning tasks. "✗" indicates cases where the solver produced an invalid solution or exceeded the allotted time limit. Speed-up refers to the speed-up in the averaged running time (on the largest dataset) achieved by ReHLine, where "$\infty$" indicates that the solver fails to solve the problem.

| TASK | DATASET | ECOS | MOSEK | SCS | DCCP | REHLINE |
|---|---|---|---|---|---|---|
| FairSVM | SPF ($\times$1e-4) | ✗ | ✗ | ✗ | ✗ | 4.25 ($\pm$0.5) |
| | Philippine ($\times$1e-2) | 1550 ($\pm$0.6) | 87.4 ($\pm$0.2) | 130 ($\pm$42) | 1137 ($\pm$9.2) | 1.03 ($\pm$0.2) |
| | Sylva-prior ($\times$1e-2) | ✗ | ✗ | ✗ | ✗ | 0.47 ($\pm$0.1) |
| | Creditcard ($\times$1e-1) | 175 ($\pm$0.2) | 64.2 ($\pm$0.1) | 161 ($\pm$405) | ✗ | 0.64 ($\pm$0.2) |
| | Fail/Succeed | 2/2 | 2/2 | 2/2 | 3/1 | 0/4 |
| | Speed-up (on Creditcard) | 273x | 100x | 252x | $\infty$ | – |

| TASK | DATASET | ECOS | MOSEK | SCS | REHLINE |
|---|---|---|---|---|---|
| ElasticQR | LD ($\times$1e-4) | ✗ | 106 ($\pm$7) | 34.9 ($\pm$25.0) | 2.60 ($\pm$0.30) |
| | Kin8nm ($\times$1e-3) | ✗ | 92.0 ($\pm$1.0) | 63.1 ($\pm$58.5) | 4.12 ($\pm$0.95) |
| | House-8L ($\times$1e-3) | 887 ($\pm$161) | 277 ($\pm$34) | ✗ | 7.21 ($\pm$1.99) |
| | Topo-2-1 ($\times$1e-2) | 4752 ($\pm$2015) | ✗ | ✗ | 3.04 ($\pm$0.49) |
| | BT ($\times$1e-0) | 7079 ($\pm$2517) | ✗ | ✗ | 2.49 ($\pm$0.56) |
| | Fail/Succeed | 3/2 | 2/3 | 3/2 | 0/5 |
| | Speed-up (on BT) | 2843x | $\infty$ | $\infty$ | – |

| TASK | DATASET | ECOS | MOSEK | SCS | HQREG | REHLINE |
|---|---|---|---|---|---|---|
| RidgeHuber | LD ($\times$1e-4) | ✗ | ✗ | ✗ | 4.90 ($\pm$0.00) | 1.40 ($\pm$0.20) |
| | Kin8nm ($\times$1e-3) | ✗ | ✗ | ✗ | 1.58 ($\pm$0.21) | 2.04 ($\pm$0.30) |
| | House-8L ($\times$1e-3) | ✗ | 925 ($\pm$2) | ✗ | 2.42 ($\pm$0.34) | 0.80 ($\pm$0.21) |
| | Topo-2-1 ($\times$1e-2) | 2620 ($\pm$1040) | 267 ($\pm$1) | 213 ($\pm$2) | 3.53 ($\pm$0.67) | 1.78 ($\pm$0.32) |
| | BT ($\times$1e-1) | ✗ | 2384 ($\pm$433) | ✗ | 12.5 ($\pm$1.8) | 5.28 ($\pm$1.31) |
| | Fail/Succeed | 4/1 | 2/3 | 4/1 | 0/5 | 0/5 |
| | Speed-up (on BT) | $\infty$ | 452x | $\infty$ | 2.37x | – |

| TASK | DATASET | ECOS | MOSEK | SCS | LIBLINEAR | REHLINE |
|---|---|---|---|---|---|---|
| SVM | SPF ($\times$1e-4) | ✗ | 372 ($\pm$1) | 237 ($\pm$27) | 12.7 ($\pm$0.1) | 3.90 ($\pm$0.10) |
| | Philippine ($\times$1e-2) | 1653 ($\pm$41) | 86.5 ($\pm$0.2) | 153 ($\pm$146) | 1.80 ($\pm$0.02) | 0.82 ($\pm$0.02) |
| | Sylva-prior ($\times$1e-3) | ✗ | 731 ($\pm$2) | 843 ($\pm$1006) | 16.0 ($\pm$0.6) | 4.08 ($\pm$0.84) |
| | Creditcard ($\times$1e-2) | 2111 ($\pm$804) | ✗ | 1731 ($\pm$4510) | 23.1 ($\pm$2.5) | 5.08 ($\pm$1.45) |
| | Fail/Succeed | 2/2 | 1/3 | 0/4 | 0/4 | 0/4 |
| | Speed-up (on Creditcard) | 415x | $\infty$ | 340x | 4.5x | – |

| TASK | DATASET | SAGA | SAG | SDCA | SVRG | REHLINE |
|---|---|---|---|---|---|---|
| sSVM | SPF ($\times$1e-4) | 39.9 ($\pm$4.6) | 28.3 ($\pm$5.0) | 15.0 ($\pm$2.4) | 41.4 ($\pm$3.9) | 4.80 ($\pm$1.20) |
| | Philippine ($\times$1e-2) | 24.3 ($\pm$27.8) | 5.53 ($\pm$9.8) | 1.47 ($\pm$0.19) | 15.8 ($\pm$6.8) | 0.89 ($\pm$0.10) |
| | Sylva-prior ($\times$1e-2) | 3.37 ($\pm$9.81) | 3.00 ($\pm$0.56) | 1.57 ($\pm$0.23) | 3.40 ($\pm$0.84) | 0.86 ($\pm$0.14) |
| | Creditcard ($\times$1e-2) | 10.4 ($\pm$1.4) | 15.0 ($\pm$2.0) | 14.0 ($\pm$1.9) | 11.2 ($\pm$1.4) | 6.36 ($\pm$1.92) |
| | Fail/Succeed | 0/4 | 0/4 | 0/4 | 0/4 | 0/4 |
| | Speed-up (on Creditcard) | 1.6x | 2.3x | 2.2x | 1.7x | – |

The results in Table 5 indicate that the proposed ReHLine algorithm/software achieves visible improvements over existing solvers. The major empirical findings are as follows. First, the amount of improvement achieved by ReHLine in terms of running time is substantial over generic solvers, including ECOS, SCS, and MOSEK based on CVX, with the largest improvement 100x-1000x speed-up on the largest-scale dataset. The improvement continues to expand even further as the scale of the problem grows. Second, ReHLine also yields considerable improvements for specialized algorithms, including LIBLINEAR in SVM, HQREG in Huber minimization, and LIGHTNING in sSVM. Third, given that the generic solvers fail or time out in multiple tasks and datasets, ReHLine has shown remarkable flexibility when it comes to solving various domain-specific problems.

# 6 Conclusion

In this paper, we present a new algorithm ReHLine designed to efficiently solve general regularized ERM with the convex PLQ loss function and optional linear constraints. Through the transformation of a PLQ loss into a composite ReLU-ReHU function, we have developed a novel CD algorithm that updates primal and dual variables simultaneously. This approach has proven to be highly effective in reducing the computational complexity while at the same time achieving a fast linear convergence rate. Based on our experiments, ReHLine demonstrates remarkable flexibility and computational efficiency, outperforming existing generic and specialized methods in a range of problems.

**Limitation.** Although our approach has exhibited substantial versatility in addressing a diverse range of problems, it is important to acknowledge that it may not be universally applicable to all convex optimization problems. The main limit of our approach stems from the restriction imposed by the formulation (1). For instance, (1) requires a convex PLQ loss function and a strict $l_2$-regularization. We recognize this limitation and intend to address it in future research pursuits. Moreover, we are interested in exploiting additional loss function properties, such as symmetry.

# 7 Acknowledgements

Ben Dai's work was supported in part by HK GRF-24302422 and GRF-14304823. Yixuan Qiu's work was supported in part by National Natural Science Foundation of China (12101389), Shanghai Pujiang Program (21PJC056), MOE Project of Key Research Institute of Humanities and Social Sciences (22JJD110001), and Shanghai Research Center for Data Science and Decision Technology. The authors confirm contribution to the paper as follows. The main algorithm: Ben Dai and Yixuan Qiu; theory and proofs: Ben Dai; C++ implementation: Yixuan Qiu; Python interface: Ben Dai and Yixuan Qiu; R interface: Yixuan Qiu; manuscript preparation: Ben Dai and Yixuan Qiu, equally.

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

# A   Example Models

**SVMs with fairness constraints (FairSVM).**   The formulation for FairSVM [48] is:

$$\min_{\boldsymbol{\beta}\in\mathbb{R}^d}\quad \frac{C}{n}\sum_{i=1}^n(1-y_i\boldsymbol{\beta}^\mathsf{T}\mathbf{x}_i)_+ + \frac{1}{2}\|\boldsymbol{\beta}\|_2^2,$$

$$\text{s.t.}\quad \frac{1}{n}\sum_{i=1}^n\mathbf{z}_i\boldsymbol{\beta}^\mathsf{T}\mathbf{x}_i\le\boldsymbol{\rho},\quad \frac{1}{n}\sum_{i=1}^n\mathbf{z}_i\boldsymbol{\beta}^\mathsf{T}\mathbf{x}_i\ge-\boldsymbol{\rho}, \tag{A.1}$$

where $\mathbf{x}_i\in\mathbb{R}^d$ is a feature vector, and $y_i\in\{-1,1\}$ is a binary label, $\mathbf{z}_i\in\mathbb{R}^{d_0}$ is a collection of centered sensitive features $(\sum_{i=1}^n\mathbf{z}_i=\mathbf{0})$, such as gender and/or race. The constraints in (A.1) constrain the correlation between the sensitive features $\mathbf{z}_i$ and the decision function $\boldsymbol{\beta}^\mathsf{T}\mathbf{x}_i$, and the constants $\boldsymbol{\rho}\in\mathbb{R}_+^{d_0}$ trade off predictive accuracy and fairness. Note that the FairSVM (A.1) can be rewritten as a ReHLine optimization of (4) with

$$\mathbf{U}\leftarrow -C\mathbf{y}^\mathsf{T}/n,\quad \mathbf{V}\leftarrow C\mathbf{1}_n^\mathsf{T}/n,\quad \mathbf{A}\leftarrow\begin{pmatrix}\mathbf{Z}^\mathsf{T}\mathbf{X}/n\\-\mathbf{Z}^\mathsf{T}\mathbf{X}/n\end{pmatrix},\quad \mathbf{b}\leftarrow\begin{pmatrix}\boldsymbol{\rho}\\\boldsymbol{\rho}\end{pmatrix},$$

where $\mathbf{1}_n=(1,\cdots,1)^\mathsf{T}$ is the length-$n$ one vector, $\mathbf{X}\in\mathbb{R}^{n\times d}$ is the feature matrix, and $\mathbf{y}=(y_1,\cdots,y_n)^\mathsf{T}$ is the response vector.

**Elastic net regularized quantile regression (ElasticQR).**   The formulation for the elastic net penalized quantile regression [19] is:

$$\min_{\boldsymbol{\beta}\in\mathbb{R}^{d+1}}\frac{1}{n}\sum_{i=1}^n\rho_\kappa(y_i-\mathbf{x}_i^\mathsf{T}\boldsymbol{\beta}_{1:d}-\beta_{d+1})+\lambda_1\|\boldsymbol{\beta}\|_1+\frac{\lambda_2}{2}\|\boldsymbol{\beta}\|_2^2, \tag{A.2}$$

where

$$\rho_\kappa(u)=u(\kappa-\mathbf{1}(u<0)) \tag{A.3}$$

is the check loss, $\kappa\in(0,1)$ is a prespecified sample quantile, $\mathbf{x}_i\in\mathbb{R}^d$ is a feature vector, $y_i\in\mathbb{R}$ is a response, and $\lambda_1,\lambda_2\ge0$ are weights of lasso and ridge penalties. Then, the ElasticQR (A.2) can be rewritten as a ReHLine optimization of (4) with

$$\mathbf{U}\leftarrow\begin{pmatrix}-\frac{\kappa}{n\lambda_2}\mathbf{1}_n^\mathsf{T} & \mathbf{0}_{d+1}^\mathsf{T}\\\frac{1-\kappa}{n\lambda_2}\mathbf{1}_n^\mathsf{T} & \mathbf{0}_{d+1}^\mathsf{T}\\\mathbf{0}_n^\mathsf{T} & \frac{\lambda_1}{\lambda_2}\mathbf{1}_{d+1}^\mathsf{T}\\\mathbf{0}_n^\mathsf{T} & -\frac{\lambda_1}{\lambda_2}\mathbf{1}_{d+1}^\mathsf{T}\end{pmatrix},\quad \mathbf{V}\leftarrow\begin{pmatrix}\frac{\kappa}{n\lambda_2}\mathbf{y}^\mathsf{T} & \mathbf{0}_{d+1}^\mathsf{T}\\-\frac{1-\kappa}{n\lambda_2}\mathbf{y}^\mathsf{T} & \mathbf{0}_{d+1}^\mathsf{T}\\\mathbf{0}_n^\mathsf{T} & \mathbf{0}_{d+1}^\mathsf{T}\\\mathbf{0}_n^\mathsf{T} & \mathbf{0}_{d+1}^\mathsf{T}\end{pmatrix},\quad \mathbf{X}\leftarrow\begin{pmatrix}\mathbf{X} & \mathbf{1}_n\\ & \mathbf{I}_{d+1}\end{pmatrix},$$

where $\mathbf{I}_{d+1}$ is the identity matrix.

**Elastic net regularized Huber minimization.**   The formulation for the elastic net penalized Huber minimization [23] is:

$$\min_{\boldsymbol{\beta}}\frac{1}{n}\sum_{i=1}^nH_\kappa(y_i-\mathbf{x}_i^\mathsf{T}\boldsymbol{\beta})+\lambda_1\|\boldsymbol{\beta}\|_1+\frac{\lambda_2}{2}\|\boldsymbol{\beta}\|_2^2, \tag{A.4}$$

where $H_\kappa(\cdot)$ is the Huber loss with a given parameter $\kappa$:

$$H_\kappa(z)=\begin{cases}z^2/2, & 0<|z|\le\kappa,\\\kappa(|z|-\kappa/2), & |z|>\kappa.\end{cases}$$

In this case, (A.4) can be rewritten as a ReHLine optimization with

$$\mathbf{S}\leftarrow\begin{pmatrix}-\sqrt{\frac{1}{n\lambda_2}}\mathbf{1}_n^\mathsf{T} & \mathbf{0}_d^\mathsf{T}\\\sqrt{\frac{1}{n\lambda_2}}\mathbf{1}_n^\mathsf{T} & \mathbf{0}_d^\mathsf{T}\end{pmatrix},\quad \mathbf{T}\leftarrow\begin{pmatrix}\sqrt{\frac{1}{n\lambda_2}}\mathbf{y}^\mathsf{T} & \mathbf{0}_d^\mathsf{T}\\-\sqrt{\frac{1}{n\lambda_2}}\mathbf{y}^\mathsf{T} & \mathbf{0}_d^\mathsf{T}\end{pmatrix},\quad \boldsymbol{\tau}\leftarrow\begin{pmatrix}\kappa\sqrt{\frac{1}{n\lambda_2}}\mathbf{1}_n^\mathsf{T} & \mathbf{0}_d^\mathsf{T}\\\kappa\sqrt{\frac{1}{n\lambda_2}}\mathbf{1}_n^\mathsf{T} & \mathbf{0}_d^\mathsf{T}\end{pmatrix},$$

$$\mathbf{U}\leftarrow\begin{pmatrix}\mathbf{0}_n^\mathsf{T} & \frac{\lambda_1}{\lambda_2}\mathbf{1}_d^\mathsf{T}\\\mathbf{0}_n^\mathsf{T} & -\frac{\lambda_1}{\lambda_2}\mathbf{1}_d^\mathsf{T}\end{pmatrix},\quad \mathbf{V}\leftarrow\mathbf{0},\quad \mathbf{X}\leftarrow\begin{pmatrix}\mathbf{X}\\\mathbf{I}_d\end{pmatrix}.$$

**Smoothed SVM (sSVM).** The formulation for the smoothed SVMs [33] is:

$$\min_{\boldsymbol{\beta}} \frac{1}{n} \sum_{i=1}^{n} V(y_i \boldsymbol{\beta}^{\mathsf{T}} \mathbf{x}_i) + \frac{1}{2}\|\boldsymbol{\beta}\|_2^2, \tag{A.5}$$

where $\mathbf{x}_i \in \mathbb{R}^d$ is a feature vector, and $y_i \in \{-1, 1\}$ is a binary label, and $V_\kappa(\cdot)$ is the modified Huber loss or the smoothed hinge loss:

$$V(z) = \begin{cases} 0, & z \geq 1, \\ (1-z)^2/2, & 0 < z \leq 1, \\ (1/2 - z), & z < 0. \end{cases}$$

In this case, (A.5) can be rewritten as a ReHLine optimization with

$$\mathbf{S} \leftarrow -\sqrt{C/n}\mathbf{y}^{\mathsf{T}}, \quad \mathbf{T} \leftarrow \sqrt{C/n}\mathbf{1}_n^{\mathsf{T}}, \quad \boldsymbol{\tau} \leftarrow \sqrt{C/n}\mathbf{1}_n^{\mathsf{T}}.$$

# B  Update Rules for (7)

## B.1  Canonical CD updates

**Update $\xi_k$ given others.** For $k = 1, \cdots, K$,

$$\xi_k^{\text{new}} = \operatorname*{argmin}_{\xi_k \geq 0} \frac{1}{2}\mathbf{a}_k^{\mathsf{T}}\mathbf{a}_k \xi_k^2 + \sum_{k' \neq k} \xi_{k'} \mathbf{a}_{k'}^{\mathsf{T}}\mathbf{a}_k \xi_k - \sum_{l,i} \lambda_{li} u_{li} \xi_k \mathbf{a}_k^{\mathsf{T}}\mathbf{x}_i - \sum_{h,i} \gamma_{hi} s_{hi} \xi_k \mathbf{a}_k^{\mathsf{T}}\mathbf{x}_i + b_k \xi_k$$

$$= \max\left( 0, -\frac{\mathbf{a}_k^{\mathsf{T}}\left(\sum_{k' \neq k} \xi_{k'}\mathbf{a}_{k'} - \sum_{l,i} \lambda_{li} u_{li}\mathbf{x}_i - \sum_{h,i}\gamma_{hi}s_{hi}\mathbf{x}_i\right) + b_k}{\|\mathbf{a}_k\|_2^2} \right). \tag{B.6}$$

**Update $\lambda_{li}$ given others.** For $l = 1, \cdots, L$ and $i = 1, \cdots, n$,

$$\lambda_{li}^{\text{new}} = \operatorname*{argmin}_{1 \geq \lambda_{li} \geq 0} \frac{1}{2}u_{li}^2 \lambda_{li}^2 \mathbf{x}_i^{\mathsf{T}}\mathbf{x}_i + \sum_{(l',i') \neq (l,i)} \lambda_{l'i'} u_{l'i'} u_{li} \mathbf{x}_{i'}^{\mathsf{T}}\mathbf{x}_i \lambda_{li}$$

$$- \sum_{k=1}^{K} \xi_k u_{li} \mathbf{a}_k^{\mathsf{T}}\mathbf{x}_i \lambda_{li} + \sum_{h',i'} u_{li}\gamma_{h'i'}s_{h'i'}\mathbf{x}_i^{\mathsf{T}}\mathbf{x}_{i'}\lambda_{li} - v_{li}\lambda_{li}$$

$$= \mathcal{P}_{[0,1]}\left( \frac{u_{li}\mathbf{x}_i^{\mathsf{T}}\left(\sum_{k=1}^{K} \xi_k \mathbf{a}_k - \sum_{(l',i') \neq (l,i)} \lambda_{l'i'}u_{l'i'}\mathbf{x}_{i'} - \sum_{h',i'}\gamma_{h'i'}s_{h'i'}\mathbf{x}_{i'}\right) + v_{li}}{u_{li}^2\|\mathbf{x}_i\|_2^2} \right). \tag{B.7}$$

**Update $\gamma_{hi}$ given others.** For $h = 1, \cdots, H$ and $i = 1, \cdots, n$,

$$\gamma_{hi}^{\text{new}} = \operatorname*{argmin}_{\tau_{hi} \geq \gamma_{hi} \geq 0} \frac{1}{2}(s_{hi}^2 \mathbf{x}_i^{\mathsf{T}}\mathbf{x}_i + 1)\gamma_{hi}^2 + \sum_{(h',i') \neq (h,i)} \gamma_{h'i'}s_{h'i'}s_{hi}\mathbf{x}_{i'}^{\mathsf{T}}\mathbf{x}_i\gamma_{hi}$$

$$- \sum_{k=1}^{K} \xi_k s_{hi} \mathbf{a}_k^{\mathsf{T}}\mathbf{x}_i \gamma_{hi} + \sum_{l',i'} s_{hi}\lambda_{l'i'}u_{l'i'}\mathbf{x}_i^{\mathsf{T}}\mathbf{x}_{i'}\gamma_{hi} - t_{hi}\gamma_{hi}$$

$$= \mathcal{P}_{[0,\tau_{hi}]}\left( \frac{s_{hi}\mathbf{x}_i^{\mathsf{T}}\left(\sum_{k=1}^{K} \xi_k \mathbf{a}_k - \sum_{(h',i') \neq (h,i)} \gamma_{h'i'}s_{h'i'}\mathbf{x}_{i'} - \sum_{l',i'} \lambda_{l'i'}u_{l'i'}\mathbf{x}_{i'}\right) + t_{hi}}{s_{hi}^2\|\mathbf{x}_i\|_2^2 + 1} \right). \tag{B.8}$$

The canonical CD updates in (B.6) - (B.8) for one full iteration require computation complexity of $\mathcal{O}\big((K + nd + nL + nH)(K + nL + nH)\big)$.

## B.2 ReHLine updates

Consider the Lagrangian function $\mathcal{L}$ in (7). Clearly,

$$\nabla_{\xi_k}\mathcal{L} = \mathbf{a}_k^\intercal \mathbf{a}_k \xi_k + \sum_{k' \neq k} \xi_{k'} \mathbf{a}_{k'}^\intercal \mathbf{a}_k - \sum_{l,i} \lambda_{li} u_{li} \mathbf{a}_k^\intercal \mathbf{x}_i - \sum_{h,i} \gamma_{hi} s_{hi} \mathbf{a}_k^\intercal \mathbf{x}_i + b_k$$

$$= \sum_{k'} \xi_{k'} \mathbf{a}_{k'}^\intercal \mathbf{a}_k - \sum_{l,i} \lambda_{li} u_{li} \mathbf{a}_k^\intercal \mathbf{x}_i - \sum_{h,i} \gamma_{hi} s_{hi} \mathbf{a}_k^\intercal \mathbf{x}_i + b_k$$

$$= \mathbf{a}_k^\intercal \boldsymbol{\beta}(\boldsymbol{\mu}) + b_k.$$

Similarly, we can derive that

$$\nabla_{\lambda_{li}}\mathcal{L} = -\big(u_{li}\mathbf{x}_i^\intercal \boldsymbol{\beta}(\boldsymbol{\mu}) + v_{li}\big), \quad \nabla_{\gamma_{hi}}\mathcal{L} = \gamma_{hi} - \big(s_{hi}\mathbf{x}_i^\intercal \boldsymbol{\beta}(\boldsymbol{\mu}) + t_{hi}\big).$$

Then the CD updates in (B.6) can be simplified as:

$$\xi_k^{\text{new}} = \mathcal{P}_{[0,+\infty)}\left(\xi_k^{\text{old}} - \frac{\nabla_{\xi_k}\mathcal{L}(\xi^{\text{old}})}{\|\mathbf{a}_k\|_2^2}\right) = \max\left(0, \xi_k^{\text{old}} - \frac{\mathbf{a}_k^\intercal \boldsymbol{\beta}^{\text{old}} + b_k}{\|\mathbf{a}_k\|_2^2}\right),$$

$$\boldsymbol{\beta}^{\text{new}} = \boldsymbol{\beta}^{\text{old}} + (\xi_k^{\text{new}} - \xi_k^{\text{old}})\mathbf{a}_k. \tag{B.9}$$

The CD updates in (B.7) are simplified as:

$$\lambda_{li}^{\text{new}} = \mathcal{P}_{[0,1]}\left(\lambda_{li}^{\text{old}} - \frac{\nabla_{\lambda_{li}}\mathcal{L}(\lambda^{\text{old}})}{u_{li}^2\|\mathbf{x}_i\|_2^2}\right) = \max\left(0, \min\left(1, \lambda_{li}^{\text{old}} + \frac{u_{li}\mathbf{x}_i^\intercal \boldsymbol{\beta}^{\text{old}} + v_{li}}{u_{li}^2\|\mathbf{x}_i\|_2^2}\right)\right),$$

$$\boldsymbol{\beta}^{\text{new}} = \boldsymbol{\beta}^{\text{old}} - (\lambda_{li}^{\text{new}} - \lambda_{li}^{\text{old}})u_{li}\mathbf{x}_i. \tag{B.10}$$

The CD updates in (B.8) are simplified as:

$$\gamma_{hi}^{\text{new}} = \mathcal{P}_{[0,\tau_{hi}]}\left(\gamma_{hi}^{\text{old}} - \frac{\nabla_{\gamma_{hi}}\mathcal{L}(\gamma^{\text{old}})}{s_{hi}^2\|\mathbf{x}_i\|_2^2 + 1}\right) = \max\left(0, \min\left(\tau_{hi}, \gamma_{hi}^{\text{old}} + \frac{s_{hi}\mathbf{x}_i^\intercal \boldsymbol{\beta}^{\text{old}} + t_{hi} - \gamma_{hi}^{\text{old}}}{s_{hi}^2\|\mathbf{x}_i\|_2^2 + 1}\right)\right),$$

$$\boldsymbol{\beta}^{\text{new}} = \boldsymbol{\beta}^{\text{old}} - (\gamma_{hi}^{\text{new}} - \gamma_{hi}^{\text{old}})s_{hi}\mathbf{x}_i. \tag{B.11}$$

The computational complexity of the new ReHLine iterations is $\mathcal{O}\big((K + nL + nH)d\big)$, as shown in (B.9) through (B.11). Notably, this complexity is *linear* with respect to both the sample size $n$ and the dimensionality $d$, making ReHLine a highly efficient algorithm.

## C  Screening rules for ReHLine

We extend the shrinking strategy in Algorithm 3 of [12] to the ReHLine solver. To begin with, the following theorem advocates utilizing "sparse" updates to optimize the performance of ReHLine.

**Theorem 4.** *Let* $\{\boldsymbol{\xi}^*, \boldsymbol{\Lambda}^*, \boldsymbol{\Gamma}^*, \boldsymbol{\beta}^*\}$ *be the convergent point of* $\{\boldsymbol{\xi}^{(q)}, \boldsymbol{\Lambda}^{(q)}, \boldsymbol{\Gamma}^{(q)}, \boldsymbol{\beta}^{(q)}\}$ *based on* (B.9)-(B.11), *then*

(A) *For* $\boldsymbol{\xi}$ *updates, let* $g_k^* = \mathbf{a}_k^\intercal \boldsymbol{\beta}^* + b_k$,

  – *if* $g_k^* > 0$, *then* $\exists q_0$ *such that* $\forall q \geq q_0$, $\xi_k^{(q)} = \xi_k^* = 0$.

(B) *For* $\boldsymbol{\Lambda}$ *updates,* $g_{li}^* = u_{li}\mathbf{x}_i^\intercal \boldsymbol{\beta}^* + v_{li}$,

  – *if* $g_{li}^* < 0$, *then* $\exists q_0$ *such that* $\forall q \geq q_0$, $\lambda_{li}^{(q)} = \lambda_{li}^* = 0$;
  – *if* $g_{li}^* > 0$, *then* $\exists q_0$ *such that* $\forall q \geq q_0$, $\lambda_{li}^{(q)} = \lambda_{li}^* = 1$.

(C) *For* $\boldsymbol{\Gamma}$ *updates, denote* $g_{hi}^* = s_{hi}\mathbf{x}_i^\intercal \boldsymbol{\beta}^* + t_{hi}$,

  – *if* $\gamma_{hi}^* = 0$ *and* $g_{hi}^* < 0$, *then* $\exists q_0$ *such that* $\forall q \geq q_0$, $\gamma_{hi}^{(q)} = 0$;
  – *if* $\gamma_{hi}^* = \tau_{hi}$ *and* $g_{hi}^* > \tau_{hi}$, *then* $\exists q_0$ *such that* $\forall q \geq q_0$, $\gamma_{hi}^{(q)} = \tau_{hi}$.

Building upon the insights from Theorem 4, we have formulated Algorithm 2 that involves shrinking certain "boundary" variables during the implementation of ReHLine.

**Algorithm 2:** ReHLine solver minimizing (1) with shrinking.

---

**Input** : $\mathbf{X} \in \mathbb{R}^{n \times d}, \mathbf{U}, \mathbf{V} \in \mathbb{R}^{L \times n}; \mathbf{S}, \mathbf{T}, \boldsymbol{\tau} \in \mathbb{R}^{H \times n}, \mathbf{A} \in \mathbb{R}^{K \times d}, \mathbf{b} \in \mathbb{R}^K$, tol $\varepsilon > 0$

1   Initialize $\boldsymbol{\xi} \geq \mathbf{0}, \mathbf{E} \geq \boldsymbol{\Lambda} \geq \mathbf{0}, \boldsymbol{\tau} \geq \boldsymbol{\Gamma} \geq \mathbf{0}$;

2   Compute $\boldsymbol{\beta} = \sum_{k=1}^{K} \xi_k \mathbf{a}_k - \sum_{i=1}^{n} \mathbf{x}_i \left( \sum_{l=1}^{L} \lambda_{li} u_{li} + \sum_{h=1}^{H} \gamma_{hi} s_{hi} \right)$;

3   Compute and store $\mathbf{r} = (r_i) \in \mathbb{R}^n, r_i = \|\mathbf{x}_i\|_2^2$, and $\mathbf{p} = (p_k) \in \mathbb{R}^K, p_k = \|\mathbf{a}_k\|_2^2$;

4   Initialize $\bar{M}_s = \infty, \bar{m}_s = -\infty, s = \{\xi, \lambda, \gamma\}, \mathcal{K} = [K], \mathcal{L} = [L] \bigotimes [n], \mathcal{H} = [H] \bigotimes [n]$;

5   **while** *maximum iteration* **do**

6      Initialize $M_s = \infty, m_s = -\infty, s = \{\xi, \lambda, \gamma\}$;

      `// CD updates for` $\boldsymbol{\xi}$

7      **for** $k \in \text{PERM}(\mathcal{K})$ **do**

8         $g_k = \mathbf{a}_k^\mathsf{T} \boldsymbol{\beta} + b_k$;

9         $(\texttt{is\_shrink}, \bar{g}_k) \leftarrow \texttt{PG\_feasible}(\xi_k, g_k, \bar{M}_\xi)$ ;

10        **if** *is_shrink* **then**

11           $\mathcal{K} \leftarrow \mathcal{K} \setminus \{k\}$; **continue**;

12        $M_\xi \leftarrow \max\{M_\xi, \bar{g}_k\}; \quad m_\xi \leftarrow \min\{m_\xi, \bar{g}_k\}$;

13        $\varepsilon^* = \max(-\xi_k, -p_k^{-1} g_k); \quad \xi_k \leftarrow \xi_k + \varepsilon^*; \quad \boldsymbol{\beta} \leftarrow \boldsymbol{\beta} + \varepsilon^* \mathbf{a}_k$;

      `// CD updates for` $\boldsymbol{\Lambda}$

14      **for** $(l, i) \in \text{PERM}(\mathcal{L})$ **do**

15         $g_{li} = -(u_{li} \mathbf{x}_i^\mathsf{T} \boldsymbol{\beta} + v_{li})$;

16         $(\texttt{is\_shrink}, \bar{g}_{li}) \leftarrow \texttt{PG\_ReLU}(\lambda_{li}, g_{li}, \bar{M}_\lambda, \bar{m}_\lambda)$;

17        **if** *is_shrink* **then**

18           $\mathcal{L} \leftarrow \mathcal{L} \setminus \{l, i\}$; **continue**;

19        $M_\lambda \leftarrow \max\{M_\lambda, \bar{g}_{li}\}; \quad m_\lambda \leftarrow \min\{m_\lambda, \bar{g}_{li}\}$;

20        $\varepsilon^* = \max(-\lambda_{li}, \min(1 - \lambda_{li}, u_{li}^{-2} r_i^{-1} g_{li})); \quad \lambda_{li} \leftarrow \lambda_{li} + \varepsilon^*; \quad \boldsymbol{\beta} \leftarrow \boldsymbol{\beta} - \varepsilon^* u_{li} \mathbf{x}_i$;

      `// CD updates for` $\boldsymbol{\Gamma}$

21      **for** $(h, i) \in \text{PERM}(\mathcal{H})$ **do**

22         $g_{hi} = \gamma_{hi} - (s_{hi} \mathbf{x}_i^\mathsf{T} \boldsymbol{\beta} + t_{hi})$;

23         $(\texttt{is\_shrink}, \bar{g}_{hi}) \leftarrow \texttt{PG\_ReHU}(\gamma_{hi}, g_{hi}, \bar{M}_\gamma, \bar{m}_\gamma)$;

24        **if** *is_shrink* **then**

25           $\mathcal{H} \leftarrow \mathcal{H} \setminus \{h, i\}$; **continue**;

26        $M_\gamma \leftarrow \max\{M_\gamma, \bar{g}_{hi}\}; \quad m_\gamma \leftarrow \min\{m_\gamma, \bar{g}_{hi}\}$;

27        $\varepsilon^* = \max(-\gamma_{hi}, \min(\tau_{hi} - \gamma_{hi}, (s_{hi}^2 r_i + 1)^{-1}(g_{hi} - \gamma_{hi})))$;

28        $\gamma_{hi} \leftarrow \gamma_{hi} + \varepsilon^*; \quad \boldsymbol{\beta} \leftarrow \boldsymbol{\beta} - \varepsilon^* s_{hi} \mathbf{x}_i$;

29      **if** $M_s - m_s \leq \varepsilon$ **and** $|M_s| \leq \varepsilon$ **and** $|m_s| \leq \varepsilon, s = \{\xi, \lambda, \gamma\}$ **then**

30        **if** $|\mathcal{K}| + |\mathcal{L}| + |\mathcal{H}| = K + n(L + H)$ **then**

31           **break**;

32        **else**

33           Take all shrunken variables back,

34           i.e., $\mathcal{K} = [K], \mathcal{L} = [L] \bigotimes [n], \mathcal{H} = [H] \bigotimes [n]$;

35           $\bar{M}_s \leftarrow \infty, \bar{m}_s \leftarrow -\infty, s = \{\xi, \lambda, \gamma\}$

36      **if** $M_s \leq 0$ **then** $\bar{M}_s \leftarrow \infty$ **else** $\bar{M}_s \leftarrow M_s, s = \{\xi, \lambda, \gamma\}$;

37      **if** $m_s \geq 0$ **then** $\bar{m}_s \leftarrow \infty$ **else** $\bar{m}_s \leftarrow m_s, s = \{\xi, \lambda, \gamma\}$;

**Output** : $\boldsymbol{\xi}, \boldsymbol{\Lambda}, \boldsymbol{\Gamma}, \boldsymbol{\beta}$

---

**Algorithm 3:** Projected gradients in ReHLine.

---

1 **Function** PG_feasible($\xi_k$, $g_k$, $\bar{M}$):
2    $\bar{g}_k \leftarrow g_k$; is_shrink $\leftarrow$ False;
3    **if** $\xi_k = 0$ **and** $g_k > \bar{M}$ **then**
4      |   is_shrink $\leftarrow$ True
5    **if** $\xi_k = 0$ **and** $g_k \geq 0$ **then**
6      |   $\bar{g}_k = 0$
   **Output:** is_shrink, $\bar{g}_k$

7 **Function** PG_ReLU($\lambda_{li}$, $g_{li}$, $\bar{M}$, $\bar{m}$):
8    $\bar{g}_{li} \leftarrow g_{li}$; is_shrink $\leftarrow$ False;
9    **if** $(\lambda_{li} = 0$ **and** $g_{li} > \bar{M})$ **or** $(\lambda_{li} = 1$ **and** $g_{li} < \bar{m})$ **then**
10      |   is_shrink $\leftarrow$ True
11    **if** $(\lambda_{li} = 0$ **and** $g_{li} \geq 0)$ **or** $(\lambda_{li} = 1$ **and** $g_{li} \leq 0)$ **then**
12      |   $\bar{g}_{li} = 0$
   **Output:** is_shrink, $\bar{g}_{li}$

13 **Function** PG_ReHU($\gamma_{hi}$, $g_{hi}$, $\tau_{hi}$, $\bar{M}$, $\bar{m}$):
14    $\bar{g}_{hi} \leftarrow g_{hi}$; is_shrink $\leftarrow$ False;
15    **if** $(\gamma_{hi} = 0$ **and** $g_{hi} > \bar{M})$ **or** $(\gamma_{hi} = \tau_{hi}$ **and** $g_{hi} < \bar{m})$ **then**
16      |   is_shrink $\leftarrow$ True
17    **if** $(\gamma_{hi} = 0$ **and** $g_{hi} \geq 0)$ **or** $(\gamma_{hi} = \tau_{hi}$ **and** $g_{hi} \leq 0)$ **then**
18      |   $\bar{g}_{hi} = 0$
   **Output:** is_shrink, $\bar{g}_{hi}$

---

## D   Technical proofs

**Lemma 1.** *Suppose $f(z)$ is a nonnegative convex PLQ with knots and extreme points $(t_k)_{k=1}^K$. Then, there exists $k_0 \in \{1, \cdots, K\}$, such that $f(z) \geq f(t_{k_0})$ for any $z \in \mathbb{R}$.*

*Proof.* We prove it by contradiction. Suppose that for there exists $z_0$ such that $f(z_0) < f(t_k)$ for all $k \in \{1, \cdots, K\}$, then the minimum can only be obtained outside $[t_1, t_K]$, that is

$$\exists z_0 < t_1, f(z_0) < f(t_k), \quad \text{or} \quad \exists z_0 > t_K, f(z_0) < f(t_k) \quad \text{for any } k \in \{1, \cdots, K\}.$$

For example, suppose that $z_0 \in (t_K, \infty)$, $f(z_0) < f(t_k)$ for any $k \in \{1, \cdots, K\}$. Note that the quadratic function in $(z_K, \infty)$ can be written as $f(z) = \frac{1}{2}f''_+(t_K)(z-t_K)^2 + f'_+(t_K)(z-t_K) + f(t_K)$. Based on the fact that $f(z_0) < f(t_K)$ and the convexity of $f(z)$, we have $f'_+(t_K) < 0$. Then, since $(t_k)_{k=1}^K$ is the set of all knots and extreme points, we have $f'(z) < 0$ for all $z > t_K$, and $\lim_{z \to \infty} f'(z) = \lim_{z \to \infty}(f''_+(t_K)(z - t_K) + f'_+(t_K)) \leq 0$. This yields that $f''_+(t_K) = 0$, alternatively, $f(z)$ is a linear function in $(t_K, \infty)$ as $f(z) = f'_+(t_K)(z - t_K) + f(t_K)$. Hence, when $z \to \infty$, $f(z) \to -\infty$, which contradicts with the fact that $f(z)$ is a nonnegative function. The same argument goes for the case when $z_0 \in (-\infty, t_1)$. This completes the proof. $\square$

**Proof of Proposition 1**

*Proof.* Note that $c\text{ReLU}(z) = c\max(z, 0) = \max(cz, 0)$, and

$$c\text{ReHU}_\tau(z) = \begin{cases} 0, & \sqrt{c}z \leq 0 \\ (\sqrt{c}z)^2/2, & 0 < \sqrt{c}z \leq \sqrt{c}\tau \\ \sqrt{c}\tau(\sqrt{c}z - \sqrt{c}\tau/2), & \sqrt{c}z > \sqrt{c}\tau \end{cases} = \text{ReHU}_{\sqrt{c}\tau}(\sqrt{c}z),$$

for $c > 0$. Then,

$$cL(pz + q) = \sum_{l=1}^{L} c\,\mathrm{ReLU}(u_l(pz + q) + v_l) + \sum_{h=1}^{H} c\mathrm{ReHU}_{\tau_h}(s_h(pz + q) + t_h)$$

$$= \sum_{l=1}^{L} \mathrm{ReLU}(cu_l(pz + q) + cv_l) + \sum_{h=1}^{H} \mathrm{ReHU}_{\sqrt{c}\tau_h}(\sqrt{c}s_h(pz + q) + \sqrt{c}t_h)$$

$$= \sum_{l=1}^{L} \mathrm{ReLU}(u_l'z + v_l') + \sum_{h=1}^{H} \mathrm{ReHU}_{\tau_h'}(s_h'z + t_h').$$

This completes the proof. $\qquad\square$

**Proof of Theorem 1**

*Proof.* The sufficiency ($\Longleftarrow$) directly follows from the definitions of ReLU and ReHU. Next, the necessity ($\Longrightarrow$) of the theorem is established using induction. Without loss of generality, for a nonnegative convex PLQ function $f(z)$, we assume that (A1) $f_+'(t_k)f_-'(t_{k+1}) \geq 0$, otherwise we can add the extreme point between $(t_k, t_{k+1}]$ into the set of knots; (A2) $\min_{k=1,\cdots,K} f(t_k) = 0$, otherwise we can add $\mathrm{ReLU}\left(\min_{k=1,\cdots,K} f(t_k)\right)$ to the final decomposition (Lemma 1 shows that the minimum of $f(z)$ can only be obtained within the knots). Next, we conduct the mathematical induction.

$\underline{K = 1.}$ Note that $f(z)$ can be rewritten as:

$$f(z) = \begin{cases} q_1^-(z) = \frac{f_-''(t_1)}{2}(z - t_1)^2 + f_-'(t_1)(z - t_1), & z \leq t_1, \\ q_1^+(z) = \frac{f_+''(t_1)}{2}(z - t_1)^2 + f_+'(t_1)(z - t_1), & z > t_1, \end{cases}$$

where $q_1^-(z)$ and $q_1^+(z)$ are left and right parts of $f(z)$ separated at $z = t_1$. According to Lemma 1 and Assumption A2, we have $f(t_1) = 0$ is the minimum of $f(z)$. Combined with Assumption A1, we have $f_-'(t) \leq 0$ for $t < t_1$, and $f_+'(t) \geq 0$ for $t > t_1$. Then, $f(z)$ can be decomposed as:

$$f(z) = \begin{cases} q_1^-(z) = \mathrm{ReHU}_\infty(-\sqrt{f_-''(t_1)}(z - t_1)) + \mathrm{ReLU}(f_-'(t_1)(z - t_1)), & z \leq t_1, \\ q_1^+(z) = \mathrm{ReHU}_\infty(\sqrt{f_+''(t_1)}(z - t_1)) + \mathrm{ReLU}(f_+'(t_1)(z - t_1)), & z > t_1. \end{cases}$$

Hence, $f(z) = \mathrm{ReHU}_\infty(-\sqrt{f_-''(t_1)}(z - t_1)) + \mathrm{ReLU}(f_-'(t_1)(z - t_1)) + \mathrm{ReHU}_\infty(\sqrt{f_+''(t_1)}(z - t_1)) + \mathrm{ReLU}(f_+'(t_1)(z - t_1))$. The necessity of the theorem for $K = 1$ then follows.

$\underline{K = K_0 + 1.}$ Assume that a convex PLQ $f \geq 0$ with $K(K \leq K_0)$ knots is ReLU-ReHU composite, now we turn to prove the statement still holds for $f(z)$ with $K_0 + 1$.

- *CASE 1.* Suppose the minimum $f(t^*) = 0$ is obtained at $\{t_2, \cdots t_{K_0}\}$ (not in the boundary). In this case, we can decompose $f(z)$ as $f(z) = f_l(z) + f_r(z)$ where $f_l(z)$ and $f_r(z)$ are defined as:

$$f_l(z) = \begin{cases} f(z), & z \leq t^*, \\ 0, & z > t^*. \end{cases} \qquad f_r(z) = \begin{cases} 0, & z \leq t^*, \\ f(z), & z > t^*. \end{cases}$$

  Both $f_l(z)$ and $f_r(z)$ are nonnegative convex PLQs with $K \leq K_0$ knots, thus can be represented as ReHUs and ReLUs.

- *CASE 2.* Suppose the minimum $f(t^*) = 0$ is obtained when $t^* = t_1$ or $t^* = t_{K_0+1}$. For example, when $t^* = t_1$, we conduct the same decomposition $f(z) = f_l(z) + f_r(z)$ at $z = t_1$, where $f_l(z)$ is a nonnegative convex PLQ with one knot, which is ReHU-ReLU composite. For $f_r(z)$, we further decompose it as:

$$f_r(z) = \mathrm{ReLU}(f_+'(t_1)(z - t_1)) + \mathrm{ReHU}_\tau\left(\sqrt{f_+''(t_1)}(z - t_1)\right) + R(z),$$

  where $\tau = \sqrt{f_+''(t_1)}(t_2 - t_1)$, and $R(z)$ (with at most $K_0$ knots) is defined as:

$$R(z) = \begin{cases} 0, & z \leq t_2, \\ f(z) - f_+'(t_1)(z - t_1) - f_+''(t_1)(t_2 - t_1)(z - (t_1 + t_2)/2), & z > t_2. \end{cases}$$

Now, we show that $R(z)$ is nonnegative convex. Note that $f(z)$ is convex, then for $z \geq t_2$,

$$
\begin{aligned}
f(z) &\geq f(t_2) + f'_-(t_2)(z - t_2) = f(t_2) + (f''_+(t_1)(t_2 - t_1) + f'_+(t_1))(z - t_2) \\
&= f'_+(t_1)(z - t_1) + f''_+(t_1)(t_2 - t_1)(z - (t_1 + t_2)/2) \\
&= \mathrm{ReLU}(f'_+(t_1)(z - t_1)) + \mathrm{ReHU}_\tau\big(\sqrt{f''_+(t_1)}(z - t_1)\big),
\end{aligned}
$$

where the first equality follows from the fact that $f(z)$ is a quadratic function in $[t_1, t_2]$, and thus $f'_-(t_2) = f'_+(t_1)(t_2 - t_1)$, and the second equality follows from $f(t_2) = \frac{f''(t_1)}{2}(t_2 - t_1)^2 + f'_-(t_1)(t_2 - t_1)$.

Thus, $R(z)$ is a nonnegative convex PLQ with $K_0$ knots $\{t_2, \cdots, t_{K_0+1}\}$, which can be decomposed as ReHUs and ReLUs. The same argument goes for the case when $t^* = t_{K_0+1}$.

This completes the proof, and the desirable results follow from the induction. $\qquad\square$

**Proof of Proposition 2**

*Proof.* The lemma follows from the fact that:

$$
\|\boldsymbol{\beta}\|_1 = \sum_{j=1}^d |\beta_j| = \sum_{j=1}^d \big( \mathrm{ReLU}(\mathbf{e}_j^\mathsf{T}\boldsymbol{\beta}) + \mathrm{ReLU}(-\mathbf{e}_j^\mathsf{T}\boldsymbol{\beta})\big),
$$

where $\mathbf{e}_j = (0, \cdots, 0, 1, 0 \cdots)$ is the $d$-tuple with all components equal to 0, except the $j$th. $\qquad\square$

**Proof of Theorem 2**

*Proof.* Note that the augmented Lagrangian of the primal problem (6) is:

$$
\begin{aligned}
\mathcal{L}_P &= \sum_{i=1}^n \sum_{l=1}^L \pi_{li} + \sum_{i=1}^n \sum_{h=1}^H \frac{1}{2}\theta_{hi}^2 + \sum_{i=1}^n \sum_{h=1}^H \tau_{hi}\sigma_{hi} + \frac{1}{2}\|\boldsymbol{\beta}\|_2^2 - \sum_{k=1}^K \xi_k(\mathbf{a}_k^\mathsf{T}\boldsymbol{\beta} + b_k) \\
&\quad - \sum_{i=1}^n \sum_{l=1}^L \lambda_{li}(\pi_{li} - u_{li}\mathbf{x}_i^\mathsf{T}\boldsymbol{\beta} - v_{li}) - \sum_{i=1}^n \sum_{h=1}^H \gamma_{hi}(\theta_{hi} + \sigma_{hi} - s_{hi}\mathbf{x}_i^\mathsf{T}\boldsymbol{\beta} - t_{hi}) \\
&\quad - \sum_{i=1}^n \sum_{l=1}^L \delta_{li}\pi_{li} - \sum_{i=1}^n \sum_{h=1}^H \psi_{hi}\sigma_{hi} \\
&= \mathbf{e}^\mathsf{T}\boldsymbol{\Pi}\mathbf{e} + \frac{1}{2}\|\boldsymbol{\Theta}\|_F^2 + \mathrm{Tr}(\boldsymbol{\tau}\boldsymbol{\Sigma}^\mathsf{T}) + \frac{1}{2}\|\boldsymbol{\beta}\|_2^2 - \boldsymbol{\xi}^\mathsf{T}(\mathbf{A}\boldsymbol{\beta} + \mathbf{b}) \\
&\quad - \mathrm{Tr}\big(\boldsymbol{\Lambda}(\boldsymbol{\Pi} - \mathbf{U}\,\mathrm{diag}(\mathbf{X}\boldsymbol{\beta}) - \mathbf{V})^\mathsf{T}\big) - \mathrm{Tr}\big(\boldsymbol{\Gamma}(\boldsymbol{\Theta} + \boldsymbol{\Sigma} - \mathbf{S}\,\mathrm{diag}(\mathbf{X}\boldsymbol{\beta}) - \mathbf{T})^\mathsf{T}\big) \\
&\quad - \mathrm{Tr}(\boldsymbol{\Delta}\boldsymbol{\Pi}^\mathsf{T}) - \mathrm{Tr}(\boldsymbol{\Psi}\boldsymbol{\Sigma}^\mathsf{T}), \tag{D.12}
\end{aligned}
$$

where $\boldsymbol{\xi} = (\xi_k) \in \mathbb{R}^K$, $\boldsymbol{\Lambda} = (\lambda_{li}) \in \mathbb{R}^{L \times n}$, $\boldsymbol{\Gamma} = (\gamma_{hi}) \in \mathbb{R}^{H \times n}$, $\boldsymbol{\Delta} = (\delta_{li}) \in \mathbb{R}^{L \times n}$ and $\boldsymbol{\Psi} = (\psi_{hi}) \in \mathbb{R}^{H \times n}$ are dual variables. The KKT conditions for the augmented Lagrangian are:

$$
\frac{\partial \mathcal{L}_P}{\partial \boldsymbol{\beta}} = \boldsymbol{\beta} - \sum_{k=1}^K \xi_k \mathbf{a}_k + \sum_{i=1}^n \mathbf{x}_i\Big(\sum_{l=1}^L \lambda_{li}u_{li} + \sum_{h=1}^H \gamma_{hi}s_{hi}\Big) = \mathbf{0},
$$

$$
\frac{\partial \mathcal{L}_P}{\partial \pi_{li}} = 1 - \lambda_{li} - \delta_{li} = 0, \quad \frac{\partial \mathcal{L}_P}{\partial \theta_{hi}} = \theta_{hi} - \gamma_{hi} = 0, \quad \frac{\partial \mathcal{L}_P}{\partial \sigma_{hi}} = \tau_{hi} - \gamma_{hi} - \psi_{hi} = 0,
$$

for all $(i, l, h)$. Then, by substituting the KKT conditions into the primal Lagrangian, the (negative) dual objective function is derived as:

$$\mathcal{L}_D = \frac{1}{2}\Big(\sum_{k=1}^{K}\sum_{k'=1}^{K}\xi_k\xi_{k'}\mathbf{a}_k^\mathsf{T}\mathbf{a}_{k'} + \sum_{l,i}\sum_{l',i'}\lambda_{li}\lambda_{l'i'}u_{li}u_{l'i'}\mathbf{x}_i^\mathsf{T}\mathbf{x}_{i'} + \sum_{h,i}\sum_{h',i'}\gamma_{hi}\gamma_{h'i'}s_{hi}s_{h'i'}\mathbf{x}_i^\mathsf{T}\mathbf{x}_{i'}\Big)$$

$$- \sum_{k=1}^{K}\sum_{l,i}\xi_k\lambda_{li}u_{li}\mathbf{a}_k^\mathsf{T}\mathbf{x}_i - \sum_{k=1}^{K}\sum_{h,i}\xi_k\gamma_{hi}s_{hi}\mathbf{a}_k^\mathsf{T}\mathbf{x}_i + \sum_{l,i}\sum_{h',i'}\lambda_{li}u_{li}\gamma_{h'i'}s_{h'i'}\mathbf{x}_i^\mathsf{T}\mathbf{x}_{i'}$$

$$+ \frac{1}{2}\sum_{h,i}\gamma_{hi}^2 + \sum_{k=1}^{K}\xi_k b_k - \sum_{l,i}\lambda_{li}v_{li} - \sum_{h,i}\gamma_{hi}t_{hi}$$

$$= \frac{1}{2}\boldsymbol{\xi}^\mathsf{T}\mathbf{A}\mathbf{A}^\mathsf{T}\boldsymbol{\xi} + \frac{1}{2}\mathrm{vec}(\boldsymbol{\Lambda})^\mathsf{T}\bar{\mathbf{U}}_{(3)}^\mathsf{T}\bar{\mathbf{U}}_{(3)}\mathrm{vec}(\boldsymbol{\Lambda}) + \frac{1}{2}\mathrm{vec}(\boldsymbol{\Gamma})^\mathsf{T}(\bar{\mathbf{S}}_{(3)}^\mathsf{T}\bar{\mathbf{S}}_{(3)} + \mathbf{I})\mathrm{vec}(\boldsymbol{\Gamma})$$

$$- \boldsymbol{\xi}^\mathsf{T}\mathbf{A}\bar{\mathbf{U}}_{(3)}\mathrm{vec}(\boldsymbol{\Lambda}) - \boldsymbol{\xi}^\mathsf{T}\mathbf{A}\bar{\mathbf{S}}_{(3)}\mathrm{vec}(\boldsymbol{\Gamma}) + \mathrm{vec}(\boldsymbol{\Lambda})^\mathsf{T}\bar{\mathbf{U}}_{(3)}^\mathsf{T}\bar{\mathbf{S}}_{(3)}\mathrm{vec}(\boldsymbol{\Gamma})$$

$$+ \boldsymbol{\xi}^\mathsf{T}\mathbf{b} - \mathrm{Tr}(\boldsymbol{\Lambda}\mathbf{V}^\mathsf{T}) - \mathrm{Tr}(\boldsymbol{\Gamma}\mathbf{T}^\mathsf{T}),$$

where $\bar{\mathbf{U}}_{(3)}$ and $\bar{\mathbf{S}}_{(3)}$ are the mode-3 unfolding of the tensors $\bar{\mathbf{U}} \in \mathbb{R}^{L\times n\times d}$ and $\bar{\mathbf{S}} \in \mathbb{R}^{H\times n\times d}$, and $\mathbf{I}$ is an identity matrix. $\qquad\square$

**Proof of Theorem 3**

*Proof.* The treatment for the proof is based on a result of coordinate descent on boxed constrained convex optimization [27]. To proceed, denote $\boldsymbol{\mu} = (\boldsymbol{\xi}^\mathsf{T}, \mathrm{vec}(\boldsymbol{\Lambda})^\mathsf{T}, \mathrm{vec}(\boldsymbol{\Gamma})^\mathsf{T})^\mathsf{T}$ as the collection of all dual variables, then the dual objective can be rewritten as:

$$\mathcal{L}_D(\boldsymbol{\mu}) = g(\mathbf{E}\boldsymbol{\mu}) + \mathbf{c}^\mathsf{T}\boldsymbol{\mu},$$

where $g(\mathbf{z}) = \frac{1}{2}\|\mathbf{z}\|_2^2$, and $\mathbf{E}$ and $\mathbf{c}$ are defined as:

$$\mathbf{E} = \begin{bmatrix} \mathbf{A}^\mathsf{T} & -\bar{\mathbf{U}}_{(3)} & -\bar{\mathbf{S}}_{(3)} \\ \mathbf{0} & \mathbf{0} & \mathbf{I}_{H\times n} \end{bmatrix} \in \mathbb{R}^{(d+Hn)\times(K+Ln+Hn)},$$

$$\mathbf{c} = \left(\mathbf{b}^\mathsf{T}, -\mathrm{vec}(\mathbf{V})^\mathsf{T}, \mathrm{vec}(\mathbf{T})^\mathsf{T}\right)^\mathsf{T} \in \mathbb{R}^{K+Ln+Hn}.$$

Now, we tend to verify assumptions of Theorem 2.1 in [27]. Specifically, (a) the dual problem is a convex box-constrained QP, thus the set of optimal solutions is nonempty; (b) $g(\mathbf{z}) = \frac{1}{2}\|\mathbf{z}\|_2^2$ is a strictly convex twice continuously differentiable function; (c) $\nabla^2 g(\mathbf{z}) = \mathbf{I}_{d+Hn}$ is always positive definite; (d) $\mathbf{E}$ has no zero column based on its definition. The desirable result then follows from Theorem 2.1 of [27]. $\qquad\square$

**Proof of Theorem 4**

*Proof.* Note that the primal problem (6) is strictly convex, thus it has the unique minimizer $\boldsymbol{\beta}^*$. Therefore, the sequence $\{\boldsymbol{\beta}^{(q)}\}_{q=1}^{\infty}$ provided by the proposed Algorithm 1 converges to the unique minimizer $\boldsymbol{\beta}^*$. Next, we check the gradients of each coordinate.

- For $\boldsymbol{\xi}$-coordinates, with $\xi_k \geq 0$,

$$g_k^{\mathrm{old}} := \nabla_{\xi_k}\mathcal{L}_D(\boldsymbol{\mu}^{\mathrm{old}}) = \mathbf{a}_k^\mathsf{T}\mathbf{a}_k\xi_k^{\mathrm{old}} + \sum_{k'\neq k}\xi_{k'}^{\mathrm{old}}\mathbf{a}_{k'}^\mathsf{T}\mathbf{a}_k - \sum_{l,i}\lambda_{li}^{\mathrm{old}}u_{li}\mathbf{a}_k^\mathsf{T}\mathbf{x}_i - \sum_{h,i}\gamma_{hi}^{\mathrm{old}}s_{hi}\mathbf{a}_k^\mathsf{T}\mathbf{x}_i + b_k$$

$$= \mathbf{a}_k^\mathsf{T}\Big(\sum_{k=1}^{K}\xi_k^{\mathrm{old}}\mathbf{a}_k - \sum_{i=1}^{n}\mathbf{x}_i\big(\sum_{l=1}^{L}\lambda_{li}^{\mathrm{old}}u_{li} + \sum_{h=1}^{H}\gamma_{hi}^{\mathrm{old}}s_{hi}\big)\Big) + b_k = \mathbf{a}_k^\mathsf{T}\boldsymbol{\beta}^{\mathrm{old}} + b_k.$$

Suppose $g_k^{(q)}$ is the gradient of $\xi_k$ based on $\boldsymbol{\mu}^{(q)}$, then $\lim_{q\to\infty}g_k^{(q)} = \mathbf{a}_k^\mathsf{T}\boldsymbol{\beta}^* + b_k =: g_k^*$. Therefore, if $g_k^* > 0$, then there exists $q_0$, for all $q \geq q_0$, we have $g_k^{(q)} > 0$, which implies that $\xi_k^{(q+1)} = 0$.

- In the same manner, for $\boldsymbol{\Lambda}$-coordinates, with $1 \geq \lambda_{li} \geq 0$,

$$g_{li}^{\text{old}} := \nabla_{\lambda_{li}} \mathcal{L}_D(\boldsymbol{\mu}^{\text{old}}) = -u_{li}\mathbf{x}_i^{\mathsf{T}}\boldsymbol{\beta}^{\text{old}} - v_{li}.$$

Then, $\lim_{q \to \infty} g_{li}^{(q)} = -u_{li}\mathbf{x}_i^{\mathsf{T}}\boldsymbol{\beta}^* - v_{li} =: g_{li}^*$,

- if $g_{li}^* < 0$, then $\exists q_0$ such that $\forall q \geq q_0$, $\lambda_{li}^{(q+1)} = \lambda_{li}^* = 0$;
- if $g_{li}^* > 0$, then $\exists q_0$ such that $\forall q \geq q_0$, $\lambda_{li}^{(q+1)} = \lambda_{li}^* = 1$.

- For $\boldsymbol{\Gamma}$-coordinates, with $\tau_{hi} \geq \gamma_{hi} \geq 0$, thus

$$g_{hi}^{\text{old}} := \nabla_{\gamma_{hi}} \mathcal{L}_D(\boldsymbol{\mu}^{\text{old}}) = \gamma_{hi}^{\text{old}} - s_{hi}\mathbf{x}_i^{\mathsf{T}}\boldsymbol{\beta}^{\text{old}} - t_{hi}.$$

Then, $\lim_{q \to \infty} g_{hi}^{(q)} = \gamma_{hi}^* - u_{li}\mathbf{x}_i^{\mathsf{T}}\boldsymbol{\beta}^* - v_{li} =: g_{hi}^*$, which implies that

- if $\gamma_{hi}^* = 0$ and $g_{hi}^* > 0$, then $\exists q_0$ such that $\forall q \geq q_0$, $\gamma_{hi}^{(q)} = 0$;
- if $\gamma_{hi}^* = \tau_{hi}$ and $g_{hi}^* < 0$, then $\exists q_0$ such that $\forall q \geq q_0$, $\gamma_{hi}^{(q)} = \tau_{hi}$.

$\square$

