# OpenReview forum: "ReHLine: Regularized Composite ReLU-ReHU Loss Minimization with Linear Computation and Linear Convergence"
_NeurIPS.cc/2023/Conference — NeurIPS 2023 poster_

### Official Review · Reviewer_gb3q · 2023-06-10

**Soundness:** 3 good
**Presentation:** 3 good
**Contribution:** 2 fair
**Rating:** 5
**Confidence:** 5

**Summary:**

This paper proposes a dual-coordinate descent solver for a class of ERM (Empricial Risk Minimization) problem with general linear inequality constraint, which is of linear convergence rate and efficient coordinate-update cost of O(n) (n is the number of samples). The main contribution is it extends existing method of dual-coordinate descent to handle problem with additional linear inequality constraints.

**Strengths:**

The paper has solid theoretical motivation, derivation and analysis, and has presented the idea clearly with solid experiments support on pratical optimization problems.

**Weaknesses:**

The paper completely missed discussion on existing works of dual-coordinate ascent on general ERM problem. There is a thread of works such as:

Shalev-Shwartz, Shai, and Tong Zhang. "Stochastic dual coordinate ascent methods for regularized loss minimization." Journal of Machine Learning Research 14.1 (2013).

on dual-coordinate ascent method for general ERM. The novelty of problem setup proposed in this reviewed paper is it adds an additional linear inequality constraints. The "piecewise-quadratic loss" is simply a special case of "smooth loss" of the above mentioned paper and SAME convergence rate applies to all this type of problem, so the author chould have made this work more general (by extending the formulation of the above work by simply adding linear inequliaty).

**Questions:**

To my knowledge, Liblinear is a special case of solver proposed in this paper, speicificaly designed for the SVM optimization problem.
Then why could solver proposed in this paper perform more efficiently than Liblinear on SVM problem? Is it implementation issue? or their objective functions are different? The author should have made this clear to avoid confusion.

**Limitations:**

The paper does not need to limit itself to the case of Relu-Rehu function.
Instead, any smooth loss (in the primal) yields strongly-convex dual problem. Therefore can apply same optimization method convergence rate proposed in the paper. (ref: https://www.jmlr.org/papers/volume14/shalev-shwartz13a/shalev-shwartz13a.pdf)

---

> ### Author Rebuttal · Authors · 2023-08-07
>
> ## Weaknesses
>
> > The paper completely missed discussion on existing works of dual-coordinate ascent on general ERM problem. There is a thread of works such as:
> >
> >> Shalev-Shwartz, Shai, and Tong Zhang. "Stochastic dual coordinate ascent methods for regularized loss minimization." Journal of Machine Learning Research 14.1 (2013).
> >
> > on dual-coordinate ascent method for general ERM. The novelty of problem setup proposed in this reviewed paper is it adds an additional linear inequality constraints. The "piecewise-quadratic loss" is simply a special case of "smooth loss" of the above mentioned paper and SAME convergence rate applies to all this type of problem, so the author chould have made this work more general (by extending the formulation of the above work by simply adding linear inequliaty).
>
> **Reply**: Thank you for bringing the reference for our attention. We will add the reference into our revision; see our global response for the discussions. In terms of novelty, we have the following important differences compared with the algorithm in the reference.
>
> 1. **Convergence rates are different**. In general, the "convex piecewise-quadratic loss" or the proposed ReHLine loss is NOT "smooth loss" but only "Lipschitz loss", thus the results in [SZ13] yielding that a sub-linear convergence (which is suboptimal compared to the linear convergence of Theorem 3 in our manuscript). Although refined analysis for "almost smooth function" is provided in [SZ13], the refined results, even for linear SVM, depend on the assumptions of underlying data distribution (see discussion in Section 5 and the definition of $N(u)$ in Theorem 16 in their paper).
> 2. **The technical details of the two methods are different**. [SZ13] and ours do share some similarities at a structual level. However, they differ in the details of the iterative steps. Specifically, [SZ13] directly solves the convex conjugate of the loss function, while our method further decomposes this loss function into simpler components for multi-step updates. Intuitively, our approach further leverages the decomposability and linearity of the loss function to perform multiple iterative updates on the decomposed components.
> 3. **Improve the practical feasibility of convex conjugate**. In fact, in each iteration, solving and implementing the convex conjugate of a general function is not an easy task, especially when the function is a composition of multiple different functions, also see the discussion in [SZ14]: "In general, this optimization problem is still not necessarily simple to solve because $\phi^*$ may also be complex." In comparison, our proposed ReLU-ReHU decomposition offers better implementation capabilities and property (e.g., Proposition 1).
> 4. As you mentioned, they cannot handle constrained optimization problems.
>
> ## Questions
>
> > To my knowledge, Liblinear is a special case of solver proposed in this paper, speicificaly designed for the SVM optimization problem. Then why could solver proposed in this paper perform more efficiently than Liblinear on SVM problem? Is it implementation issue? Or their objective functions are different? The author should have made this clear to avoid confusion.
>
> **Reply**: Thank you for the comments. It is indeed correct that the algorithm used in LibLinear to solve SVM is a special case of our method. We make this claim in order to highlight the efficiency of our implementation/software, as we regard software as a significant contribution of our work. We will revise the manuscript to clarify the confusion that this advantage is caused by implementation.
>
> ## Limitations:
>
> > The paper does not need to limit itself to the case of Relu-Rehu function. Instead, any smooth loss (in the primal) yields strongly-convex dual problem. Therefore can apply same optimization method convergence rate proposed in the paper. (ref: https://www.jmlr.org/papers/volume14/shalev-shwartz13a/shalev-shwartz13a.pdf)
>
> **Reply**: Thank you for the comments. As we mentioned in the previous point, our optimization objective is not a smooth function, and therefore does not readily yield linear convergence results based on the theoretical results in [SZ13]. In addition to theoretical considerations, we also place great emphasis on the algorithm's computational efficiency. Specifically, in the proposed framework, each coordinate can be solved with a simple analytic solution. This feature has demonstrated excellent efficacy and practical feasibility in our experiments.
>
> [SZ13] Shalev-Shwartz and Zhang (2013). Stochastic Dual Coordinate Ascent Methods for Regularized Loss Minimization.
>
> [SZ14] Shalev-Shwartz and Zhang (2014). Accelerated Proximal Stochastic Dual Coordinate Ascent for Regularized Loss Minimization.

---

### Official Review · Reviewer_2E2D · 2023-07-05

**Soundness:** 4 excellent
**Presentation:** 3 good
**Contribution:** 4 excellent
**Rating:** 7
**Confidence:** 3

**Summary:**

This paper proposes a new optimization algorithm for convex piecewise linear-quadratic objectives with $L_2$ regularization and linear constraints, which achieves the best known iteration complexity simultaneously with the best known per-iteration computational cost, resulting in a smaller total computational complexity than any previous algorithm. Experiments show that the proposed algorithm achieves ~1000x speedup compared to previous solvers and improves over problem-specific solvers despite its generality.

**Strengths:**

1. The considered optimization problem is broad, encompassing training for many linear models (SVM, LAD, SVR, QR) and constraint sets, including fairness.
2. The technique appears to be novel.
3. The comparison of total complexity (Table 2) demonstrates a significant theoretical advantage over previous algorithms.
4. The empirical comparison (Table 5) shows that the proposed algorithm enjoys a significant practical advantage over previous algorithms.

**Weaknesses:**

1. The experimental evaluation is missing a few details that need clarifying. For example, line 212 states that the BenchOpt framework is used "to implement optimization benchmarks for all the SOTA solvers". I find this statement a little unclear. Did the authors implement all of the baseline solvers from scratch? If not, what do they mean by "implement optimization benchmarks"? Also, is it common to implement these solvers in Python? My understanding is that the production solvers are most commonly implemented in C/C++, and using a slower language like Python might amplify the difference in speed between algorithms.

**Questions:**

1. Line 123 states that the proposed algorithm is inspired by LibLinear, but there is no further discussion of the relationship between ReHLine and LibLinear. What components of ReHLine are inspired by LibLinear, and what components of ReHLine are novel compared to LibLinear?
2. The introduction mentions that this work focuses on the case that the dataset size is much larger than the input dimension and the number of constraints, but this large-scale condition is not discussed further. Why is this large-scale condition significant to your approach, and how does ReHLine compare to baseline algorithms when $n$ is small?
3. Please clarify the questions I posed about the details of experimental evaluation in the Weaknesses section.

**Limitations:**

The authors provided a reasonable discussion of the limitations of their approach. A discussion of potential negative societal impact is, in my opinion, not necessary for this work.

---

> ### Author Rebuttal · Authors · 2023-08-07
>
> ## Weaknesses
>
> > The experimental evaluation is missing a few details that need clarifying. For example, line 212 states that the BenchOpt framework is used "to implement optimization benchmarks for all the SOTA solvers". I find this statement a little unclear. Did the authors implement all of the baseline solvers from scratch? If not, what do they mean by "implement optimization benchmarks"?
>
> **Reply**: We apologize for the confusion. To clarify, we would like to mention that we do not develop baseline  solvers from scratch in our study. Instead, we utilize well-established  software for benchmarking purposes. We appreciate your suggestion and will revise the manuscript accordingly in order to clarify any potential  misleading description.
>
> > Also, is it common to implement these solvers in Python? My understanding is that the production solvers are most commonly implemented in C/C++, and using a slower language like Python might amplify the difference in speed between algorithms.
>
> **Reply**: Thank you for the thorough comments. All the methods that we compared, including our own, are fundamentally built upon C/C++ backends. Python just serve as a facilitator by providing an API interface that enhances their usability. See more description of backends in CVXPY and CVXCORE. Python acts as a bridge between the users and the underlying C/C++ backends, enabling a seamless and more user-friendly experience. Therefore, our Python experiments do not significantly impact the execution speed of the software itself. As a result, the experimental results are reliable and reproducible. We will add more descriptions in our manuscript.
>
> ## Questions
>
> > Line 123 states that the proposed algorithm is inspired by LibLinear, but there is no further discussion of the relationship between ReHLine and LibLinear. What components of ReHLine are inspired by LibLinear, and what components of ReHLine are novel compared to LibLinear?
>
> **Reply**: The development of ReHLine was largely inspired by the linear computational complexity and linear convergence property of LibLinear, which contribute to the great success of LibLinear in practice. However, in LibLinear, these properties are developed specifically ONLY for SVMs.
>
> One major merit of ReHLine is that these good properties actually apply to a much broader loss function class (with or without constraints), characterized by the convex PLQ functions. Therefore, with ReHLine, the efficiency of LibLinear can be extended to many other ERM problems, and LibLinear can be viewed as a special case of ReHLine.
>
> > The introduction mentions that this work focuses on the case that the dataset size is much larger than the input dimension and the number of constraints, but this large-scale condition is not discussed further. Why is this large-scale condition significant to your approach, and how does ReHLine compare to baseline algorithms when $n$ is small?
>
> **Reply**: As is shown in Table 1, one major contribution of this article is to reduce the per-iteration cost from $\mathcal{O}(n^2)$ to $\mathcal{O}(n)$, so that ReHLine demonstrates great advantages when $n$ is large and forms the bottleneck of computation. It can be seen that equation (8) is a quadratic function of $(K+nL+nH)$ variables. If $n$ is small, then the scale of the quadratic programming problem (7) is also small, which means that a general-purpose solver may already be sufficient.

---

> > ### Comment · Reviewer_2E2D · 2023-08-18
> >
> > Thank you for your response. I am satisfied with the quality of the paper and your answers cleared a few details up, therefore I will keep my rating the same.

---

### Official Review · Reviewer_xA17 · 2023-07-23

**Soundness:** 3 good
**Presentation:** 3 good
**Contribution:** 3 good
**Rating:** 6
**Confidence:** 2

**Summary:**

This paper studies the Empirical Risk Minimization(ERM), which is an important framework of machine learning, and focus on a general regularized ERM based on a convex PLQ loss with linear constraints. According to the authors, the existing algorithms are faced with the problem of slow convergence or high computational cost. The authors leverage the linear property of Karush-Kuhn-Tucker conditions and propose ReHLine, whose total complexity is lower than existing algorithms.


## Comments
It is recommended to indicate the unit of running time on Table 5.

**Strengths:**

1. The authors cleverly decompose convex PLQ into a series of ReLU and the so-called rectified Huber units to deal with the problem that the standard definition of convex PLQ is not convenient to optimization algorithms.When solving the decomposed optimization problem, the authors prove that their ReHLine algorithm can achieve linear convergence rate.
2. After comparing with other state-of-the-art algorithms on multiple datasets, the experimental results show that the ReHLine algorithm can solve a variety of domain-specific problems, presenting excellent flexibility, and on large-scale datasets, the algorithm shows excellent acceleration ratio.

**Weaknesses:**

Actually, rather than those solvers tested in the experiment part, there are a number of general optimization solvers, including CPLEX, GUROBI, SCIP, etc. Those optimization solvers have exhibited great success in solving linear/nonlinear continuous/discrete optimization problems. Have you compared your proposed method against those general optimization solvers?

**Questions:**

Please see my question in the "Weakness" section.

**Limitations:**

Not applicable.

---

> ### Author Rebuttal · Authors · 2023-08-07
>
> ## Weaknesses
>
> > Actually, rather than those solvers tested in the experiment part, there are a number of general optimization solvers, including CPLEX, GUROBI, SCIP, etc. Those optimization solvers have exhibited great success in solving linear/nonlinear continuous/discrete optimization problems. Have you compared your proposed method against those general optimization solvers?
>
> **Reply**: Thank you for drawing our attention to the availability of more general solvers. Solving Constraint Integer Programs (SCIP) is specifically designed for integer programming, so it does not align well with the nature of our problem. CPLEX and GUROBI are commercial solvers which require licenses to run. At present, we only obtain free licenses for conducting small-scale experiments, and they have failed for running the datasets described in our paper. To illustrate their effectiveness, we run the solvers in simulated datasets with RidgeQR compared with all other solvers (`fail` indicates that we are unable to obtain results under the current licenses). As indicated in the following table, the proposed algorithm continues to exhibit remarkable superiority over CPLEX and GUROBI.
>
> | RidgeQR | CPLEX | ReHLine | GUROBI | MOSEK | SCS | ReHLine |
> |:------|----------:|----------:|----------:|----------:|----------:|----------:|
> |n=100  | 5.073E-02 | 7.370E-05 | 1.337E-02 | 8.348E-03 | 2.082E-03 | 1.322E-04 |
> |n=300  | 1.437E-01 | 8.979E-05 | fail      | 1.024E-02 | 1.207E-02 | 4.093E-04 |
> |n=500  | fail      | 1.091E-04 | fail      | 1.260E-02 | 1.952E-02 | 4.160E-04 |
> |n=700  | fail      | 1.365E-04 | fail      | 1.522E-02 | 2.865E-02 | 5.973E-04 |
> |n=1000 | fail      | 1.870E-04 | fail      | 1.828E-02 | 2.663E-02 | 1.459E-03 |

---

> > ### Comment · Reviewer_xA17 · 2023-08-16
> > **Thanks for the response**
> >
> > Thanks for the response! The authors have addressed my comment.

---

### Official Review · Reviewer_7RWz · 2023-07-24

**Soundness:** 3 good
**Presentation:** 3 good
**Contribution:** 2 fair
**Rating:** 5
**Confidence:** 3

**Summary:**

This paper introduces an algorithm, ReHLine, which has a linear convergence rate on minimizing convex piecewise linear-quadratic loss functions. Experiments on several tasks show great performance gains over existing algorithms.

**Strengths:**

General: The paper is clean well-written and easy to follow. Definitions are Proposition are well explained with examples.


Pros:

* Introduces ReLU-ReHU decomposition and show commonly-used loss functions have their ReLU-ReHU representation.
* Finds the dual formulation and shows that proposed algorithm has linear convergence rate by Coordinates Descent.
* Experimental results greatly outperform existing algorithms.

Cons:

* The main contribution of this paper is proposal of ReLU-ReHU decomposition and its resulting formulation. Its convergence result is shown by classical result. I feel the contribution is limited and marginally above the borderline.


In all, this paper is well written and the framework ReLU-ReHU is promising. Although the contribution seems limited but I am happy to vote for accept.

----------

Main body checked but did not throughly go through appendix.

**Weaknesses:**

See above.

**Questions:**

See above.

**Limitations:**

See above.

---

> ### Author Rebuttal · Authors · 2023-08-07
>
> ## Weaknesses
>
> > The main contribution of this paper is proposal of ReLU-ReHU decomposition and its resulting formulation. Its convergence result is shown by classical result. I feel the contribution is limited and marginally above the borderline.
>
> **Reply**: Thanks for the comments. We would like to point out that our contribution has different aspects, and the convergence result is only one of them. As can be seen in Table 1, there exist other algorithms that achieve the same convergence rate as ReHLine. However, the total computational cost is affected by both the convergence rate and the per-iteration cost. Making one of the components efficient is possible, but achieving **both** is nontrivial.
>
> To make the linear convergence and linear computational complexity **simultaneously hold**, we need a special relation between the primal and dual variables, as is characterized by equation (9), and the objective function needs to have some specific structures to support the convergence rate. One major finding in this article is that the relation (9) is met by a very general loss function class, the convex PLQ functions, and the resulting algorithm has provable linear convergence. To the best of our knowledge, prior to ReHLine, these properties were only studied for SVMs in the LibLinear solver, and now we have greatly expanded the class of models that enjoy the linear convergence and linear computation properties.

---

### Official Review · Reviewer_kbfD · 2023-07-25

**Soundness:** 3 good
**Presentation:** 3 good
**Contribution:** 2 fair
**Rating:** 6
**Confidence:** 3

**Summary:**

The authors consider tackling the optimization fo finite sum of piecewise linear quadratic functions (PLQ) arising from e.g. robust empirical risk minimization by means of a reformulation of PLQ functions as sums of ReLU and smoothed ReLU functions combined with a stochastic dual ascent algorithm. The authors start by presenting how any PLQ function can be formulated as sums of of ReLU and smoothed ReLU functions with a table summarizing some common ones. Then the proposed algorithm is presented whose convergence is ensured by previous work. Experimental results advocate for the superiority of the approach compared to old all-purpose solvers in various classification tasks.

**Strengths:**

- The generality of the approach is well defended by the flexibility of the parameterization with ReLU and smoothed ReLU functions.
- The proposed algorithm captures efficiently the underlying structure of the algorithm. In particular, compared to primal-based approaches (which could easily be implemented using the proximal operators of the losses), the approach can handle constraints.
- The proposed algorithm is hyper-parameter free.
- The experiments could be a strength, although they are simply misleading in their current state.

**Weaknesses:**

- The paper lacks some important related work. In particular, the proposed algorithm appears to simply be a form of stochastic coordinate dual ascent. Such a note would help gain perspective on the contributions of the authors.
- A
- Numerous algorithms have been developed to tackle empirical risk minimization problems. For example, s-SVM or SVM^2 losses are smooth and could be tackled by fast incremental solvers such as SAGA, SVRG or SDCA or even their accelerated versions such as [1]. see also [2]. Non-smooth losses can also generally easily be smoothed, see e.g. [3], to be amenable theoretically and in practice to fast resolution by fast incremental algorithms. True, most of these algorithms require stepsizes, which make the proposed approach friendlier at first. However, SDCA for example does not require any and could a priori be formulated like the proposed algorithm (using that the proximal operator of the conjugate of the losses is available here). A thorough discussion and comparison to all of the aforementioned algorithms is necessary to held this paper to the claims the authors make. Additional
- Non-negativity, box and monotonicity constraints can be handled by simple projections and primal algorithms could also be considered in those cases.
- The current approach cannot handle non-strongly convex penalty such as sparsity inducing penalties.
- The experimental results and claims are misleading: no one uses all-purpose solvers such as MOSEK. In particular no one uses interior point methods for empirical risk minimization. Claiming 1000 times improvement is overselling the method which in turns is detrimental to the contributions of the paper. Usual comparisons consist in claiming improvements over state of the art method, which appear to hold but at much smaller scale. The proposed algorithm may provide quantitative benefits, but a comprehensive comparison would help the paper.

Minor details:
- E appears not have been defined in Theorem 2. Similarly recalling what a mode-3 unfolding of a tensor is, would improve the readability of the paper.

[1] A. Defazio. A Simple Practical Accelerated Method for Finite Sums. NIPS 2016
[2] S Shalev-Shwartz, T Zhang. Accelerated Mini-Batch Stochastic Dual Coordinate Ascent. NIPS 2013
[3] A Beck. Smoothing and First Order Methods: A Unified Framework. SIOPT 2012

**Questions:**

- Can the authors do comparisons with state of the art algorithms for empirical risk minimization for each of the proposed problems rather than using all-purpose algorithms? A list of potential candidates has been given in the weakness section.
- In particular, can the authors discuss and compare empirically their algorithm against SDCA in the absence of constraints?
- How can the proposed algorithms handle nonlinear models such as kernel methods?

**Limitations:**

- No support for sparsity inducing norms

---

> ### Author Rebuttal · Authors · 2023-08-07
>
> ## Weaknesses
>
> > (related work)
>
> **Reply**: Thanks for the comments. We emphasize that our contribution is not only applying CD (and its extensions such as SDCA) to a specific optimization problem, but also to understand the class of problems such that the CD variant can be implemented with linear computational complexity and provable linear convergence.
>
> In our revised manuscript, we have added more related works and discussed their connections with ReHLine. See our global response.
>
> > (s-SVM example)
>
> **Reply**: Following your suggestion, we have now added a new benchmark for smoothed SVM (s-SVM) to compare the running times of various algorithms, including SAGA, SAG, SDCA, SVRG, and the proposed ReHLine. In this experiment, the specialized implementations of the algorithms for s-SVM are implemented using the Python library `lightning`. It is worth mentioning that computationally demanding parts in `lightning` are implemented via `Cython` (C-Extensions for Python), so we believe that the comparison is roughly fair.
>
> |s-SVM|SAGA|SAG|SDCA|SVRG|ReHLine|
> |-|-|-|-|-|-|
> |SPF|3.154E-03|2.462E-03|1.316E-03|3.095E-03|5.298E-04|
> |philippine|1.292E-01|7.322E-02|2.210E-02|1.325E-01|1.086E-02|
> |sylva_prior|5.084E-02|4.055E-02|2.371E-02|3.140E-02|1.008E-02|
> |creditcard|1.146E-01|1.625E-01|1.438E-01|1.164E-01|6.451E-02|
> |**speed-up**|1.8~11.9x|2.5~6.7x|2.0~2.5x|1.8~12.2x|--|
>
> As indicated in the table, even in smooth problems like s-SVM, ReHLine has shown reasonable improvement (or at least demonstrated comparable performance) of these specialized implementations.
>
> > (other algorithms for ERM)
>
> **Reply**: Thanks for providing these references. As we have shown in the response to the first question, in the planned revision we have added discussions on such related works. Some key points are again summarized below:
>
> 1. SAG, SVRG, and SAGA require smooth loss functions, with an optional non-smooth term that has an easy proximal operator.
> 2. Smoothing methods as in [BT12] do not have a linear convergence guarantee.
> 3. ReHLine can handle linear constraints, whereas SDCA does not consider such cases.
> 4. SDCA requests the convex conjugate of the loss function, but computing the convex conjugate of a general convex PLQ function is not straightforward.
> 5. SDCA is only guaranteed a sub-linear convergence rate for **non-smooth** losses.
>
> > (special constraints)
>
> **Reply**: Of course, these methods can make use of the special structures of the constraints, but in ReHLine all linear constraints can be handled in a unified way, without sacrificing the computational efficiency and convergence rate.
>
> > (sparsity inducing penalties)
>
> **Reply**: Thank you for bringing this up. While strong convexity is necessary for our theoretical analysis, sparsity can still be achieved by incorporating elastic net penalties (L1+L2), as described in Proposition 2. Additionally, for L1 penalty, the technique used in Equation (5) of Section 4.2 in [SZ14] can also be applied to ReHLine.
>
> > (experimental results)
>
> **Reply**: Thank you for the comments. Following your suggestion, we have added a new benchmark to compare our proposed method with other specialized implementations in s-SVM. Please refer to our reply to the previous points for more details.
>
> However, we would like to emphasize that there is still a certain usefulness for all-purpose solvers. For example, even in simple problems like FairSVM [39, 40], the authors opted to use an all-purpose solver but less efficient software such as DCCP for implementation.
>
> Since our algorithm possesses some attributes similar to all-purpose solvers, including custom losses and constraints, to clarify our experimental conclusions, we would like to present them in two parts: (i) When compared to all-purpose solvers, ReHLine has achieved significant improvements; (ii) In the case of specialized algorithms/implementations, ReHLine can compete with and even outperfom specialized implementations in specific scenarios.
>
> > (minor details)
>
> **Reply**: Thanks for pointing out. $\mathbf{E}$ stands for a matrix of all ones, and the mode-3 unfolding of a tensor $\mathbf{U}=(u_{ijk})\in\mathbb{R}^{m\times n\times l}$ is the matrix $V=(v_{ks})\in\mathbb{R}^{l\times (mn)}$, where the $k$-th row of $V$ is the vectorization of the $k$-th slice of $\mathbf{U}$, i.e., $V_{k\cdot}=\mathbf{vec}(U_{\cdot\cdot k})^T$.
>
> ## Questions
>
> > (first question already covered in the Weakness section)
>
> > (comparison with SDCA)
>
> **Reply**: The differences between ReHLine and SDCA are summarized in the global response.
>
> > (kernel methods)
>
> **Reply**: Thank you for the insightful comments. Our method can be extended to the following kernel learning problem:
>
> $$\min_{\beta}\sum_{i=1}^n L_i(\beta^TK_i)+\frac{1}{2}\|\beta\|_K^2,\quad\text{s.t. }A\beta+b\geq 0,$$
>
> where $K$ is a p.d. kernel matrix, $\beta$ becomes a length-$n$ vector, and the prediction function is now $f(x_j)=\beta^T K_j$.
>
> Using Cholesky factorization $K=Q^TQ$, and denoting $\alpha=Q\beta$, it can be rewritten as:
>
> $$\min_{\alpha}\sum_{i=1}^n L_i(K^T_i Q^{-1}\alpha)+\frac{1}{2}\|\alpha\|_{2}^2,\quad\text{s.t. }AQ^{-1}\alpha+b\geq 0,$$
>
> which follows the form of (1) in our paper, with $x_i \leftarrow K^T_i Q^{-1}$, $A\leftarrow AQ^{-1}$, thus can be solved by the proposed ReHLine method.
>
> However, one major contribution of this article is to reduce the per-iteration cost from $\mathcal{O}(n^2)$ to $\mathcal{O}(nd)$. In the kernel learning $d = n$, yielding the computational complexity re-increase to $\mathcal{O}(n^2)$. In this light, when it comes to solving problems related to kernel learning or high-dimensional problems where d is greater than n, we do not recommend using the ReHLine algorithm. We will add more discussion in the revision.
>
> [BT12] Beck and Teboulle (2012). Smoothing and First Order Methods: A Unified Framework.
>
> [SZ14] Shalev-Shwartz and Zhang (2014). Accelerated Proximal Stochastic Dual Coordinate Ascent for Regularized Loss Minimization.

---

> > ### Comment · Reviewer_kbfD · 2023-08-14
> > **Thanks for the answer**
> >
> > I thank the authors for their detailed answers. I appreciate the additional comparisons. I believe the paper will highly benefit from such nuanced comparisons. Highlighting the versatility of the approach while remaining competitive in comparison to previous dedicated algorithms is, I believe, a better way to present this work. I increased my score in light of the more detailed discussion of previous methods.

---

> > > ### Author Response · Authors · 2023-08-17
> > >
> > > Thank you for the insightful comments. We are grateful for the increased rating to our paper. We will further revise our manuscript based on your suggestions.

---

### Official Review · Reviewer_xSEV · 2023-07-25

**Soundness:** 3 good
**Presentation:** 3 good
**Contribution:** 3 good
**Rating:** 7
**Confidence:** 3

**Summary:**

This paper introduced a new function class called composite ReLU-ReHU which is shown to be equivalent to the class of convex PLQ functions. Based on the ReLU-ReHU decomposition of convex PLQ functions, the authors then formulated a new box-constrained quadratic programming optimization problem, named ReHLine optimization which can be solved efficiently by the ReHLine algorithm. The proposed solver has a provable linear convergence rate and a linear per-iteration computation complexity. Benchmarking on various tasks and datasets, the ReHLine showed significant improvement over both generic and specific solvers, especially on large-scale datasets.

**Strengths:**

- This paper is well-written and easy to follow.
- To the best of my knowledge, the composite ReLU-ReHU functions and the ReHLine algorithm are novel.
- The effectiveness of the proposed method is adequately backed by both theoretical results and empirical results.
- The ReHLine algorithm successfully tackles any convex PLQ loss functions, thus it shows great potential for many machine learning tasks.


**Weaknesses:**

- While authors claimed that the simplification of Canonical CD updates is highly non-trivial, it seems to be a straight application of [18] to the ReHLine optimization problem.

**Questions:**

1. Are the experimental results using Algorithm 1 or Algorithm 2 in the Appendix? Could the authors provide the running time of both algorithms for comparison?
2. Could the authors provide an ablation study to demonstrate the objective gaps of different solvers when increasing the number of iterations?

Minor:
- There is a typo in Proposition 2: $T \leftarrow \begin{pmatrix} \sqrt{\frac{2}{\lambda_2}} \boldsymbol{T} & \boldsymbol{0}_d^T \end{pmatrix}$.

**Limitations:**

The authors have discussed the limitations of their work in Section 6.

---

> ### Author Rebuttal · Authors · 2023-08-07
>
> ## Weaknesses
>
> > While authors claimed that the simplification of Canonical CD updates is highly non-trivial, it seems to be a straight application of [18] to the ReHLine optimization problem.
>
> **Reply**: Thanks for the comments. When the ReHLine optimization problem (1) and (3) is given, the paper seems to be a direct application of LibLinear [18]. However, the main contribution of our paper is actually the induction/construction of a class of problems that allows for the generalization of the LibLinear [18] algorithm, achieving both linear convergence and linear computational complexity.
>
> Specifically, we would like to point out that whether the canonical CD can be simplified largely depends on whether the special relationship between the primal and dual variables as in equation (9) exists, and meanwhile we request the algorithm to have linear convergence. [18] is an important reference as it discovers this relationship in SVMs and proves its linear convergence. However, to the best of our knowledge, prior to ReHLine it is unknown whether these two properties hold for other more general cases. In this article, we have greatly expanded the class of models that possess these properties. In particular, we show that they are available not only for the hinge loss as in SVMs, but also for all convex PLQ functions. We think that this generalization is novel and non-trivial.
>
> ## Questions
>
> > Are the experimental results using Algorithm 1 or Algorithm 2 in the Appendix? Could the authors provide the running time of both algorithms for comparison?
>
> **Reply**: Thank you for the comments. We are using Algorithm 2 (ReHLine solver with shrinking) for the experiments. Following your suggestion, we additionally conducted experiments to compare the ReHLine with and without shrinkage in the following tables.
>
> The experimental results suggest that our algorithm's superiority (compared to other methods) is not influenced by shrinkage. In fact, in most cases, ReHLine without shrinkage may even outperform the one with shrinkage.
>
> | FairSVM | ReHLine(shrink=False) | ReHLine(shrink=True) |
> |:-----------|----------:|----------:|
> |SPF         | 3.951E-04 | 4.130E-04 |
> |philippine  | 1.186E-01 | 1.602E-02 |
> |sylva_prior | 9.234E-03 | 1.010E-02 |
> |creditcard  | 3.318E-01 | 2.157E-01 |
>
>
> | ElasticQR | ReHLine(shrink=False) | ReHLine(shrink=True) |
> |:---------------|----------:|----------:|
> |liver-disorders | 8.111E-05 | 2.165E-04 |
> |kin8nm          | 6.109E-04 | 3.491E-03 |
> |house_8L        | 1.618E-03 | 9.481E-03 |
> |topo_2_1        | 8.242E-03 | 4.549E-02 |
> |BT              | 1.454E-01 | 2.918E+00 |
>
>
> | SVM | ReHLine(shrink=False) | ReHLine(shrink=True) |
> |:-----------|----------:|----------:|
> |SPF         | 8.351E-04 | 3.968E-04 |
> |philippine  | 1.193E-01 | 1.083E-02 |
> |sylva_prior | 4.841E-03 | 5.709E-03 |
> |creditcard  | 2.973E-02 | 5.842E-02 |
>
>
> | ElasticHuber | ReHLine(shrink=False) | ReHLine(shrink=True) |
> |:---------------|----------:|----------:|
> |liver-disorders | 1.121E-04 | 1.210E-04 |
> |kin8nm          | 8.201E-04 | 2.176E-03 |
> |house_8L        | 7.009E-04 | 1.004E-03 |
> |topo_2_1        | 1.025E-1  | 2.084E-02 |
> |BT              | 1.737E-01 | 4.263E-01 |
>
>
> > Could the authors provide an ablation study to demonstrate the objective gaps of different solvers when increasing the number of iterations?
>
> **Reply**: Following your suggestion, we have included the figures depicting the optimization progress, specifically the objective function value over time or step, for all the benchmarks. Kindly refer to the attached PDF document for a comprehensive overview.
>
> > Minor: There is a typo in Proposition 2: $\mathbf{T} \leftarrow\left(\begin{array}{cc}\sqrt{\frac{2}{\lambda_2}} \mathbf{T} & \mathbf{0}^\intercal_d\end{array}\right)$
>
> **Reply**: Thanks for pointing out the typo. We will fix it in the revision.

---

> > ### Comment · Reviewer_xSEV · 2023-08-14
> > **Respond to Authors**
> >
> > I thank the authors for their response. After reading their rebuttal, I appreciate that the authors have addressed all of my concerns.

---

### Author Rebuttal · Authors · 2023-08-07

# To All Reviewers

Thank you all reviewers for the encouraging and insightful comments. We appreciate the time and effort the reviewers have dedicated to providing valuable feedback on our manuscript. In this round, we have made every effort to address all the comments of the reviewers. **The point-to-point responses are provided in the reply**.

Additionally, we would like to take this opportunity to highlight our contributions in software development. We have dedicated a significant amount of time to optimize our C/C++ implementation, Python API, quick routines of various losses and constraints, and provide comprehensive documentation. Our motivation behind these efforts stems from the fact that there are currently not many practical software solutions available. For instance, in simple problems like FairSVM [39,40], the authors opted to use generic but less efficient software such as DCCP for implementation. Given that our software not only operates efficiently but also offers a high level of flexibility and practicality in various applications, we sincerely hope that the reviewers will take into consideration our contribution to software when evaluating this article.

As suggested by the reviewers, one point we want to highlight globally is the added discussions on the existing literature. In our revised manuscript, we have included more related works and discussed their connections with ReHLine. Below is an excerpt of the planned revision:

> There are multiple existing works developed to tackle ERM problems, such as SAG, SVRG, SAGA, and SDCA. However, SAG, SVRG, and SAGA can only handle smooth loss functions, with an optional non-smooth term that has an easy proximal operator. SDCA applies to general loss functions, but it requests the convex conjugate of loss functions, which is not necessarily simple to compute. Moreover, it only guarantees a sub-linear convergence rate for non-smooth loss functions. In contrast, ReHLine supports all convex PLQ loss functions with optional general linear constraints, which are non-smooth by construction, and they all enjoy a linear computational complexity and provable linear convergence.

>
> For non-smooth loss functions, another existing method to solve ERM is the smoothing technique (Beck and Teboulle, 2012), which approximates the non-smooth terms by smooth functions, and then uses gradient-based algorithms to solve the smooth problem. However, the choice of the smoothing function and smoothing parameter typically requires additional knowledge, and the linear convergence rate is not always guaranteed.

Furthermore, we would like to supplement the following important differences between ReHLine and SDCA.

1. **Convergence rates are different for non-smooth losses**. In general, the PLQ or the proposed ReHLine loss is non-smooth but Lipschitz function, thus the theoretical results of SDCA [SZ13] yield a *sub-linear* convergence (which is suboptimal compared to the linear convergence of Theorem 3 in our manuscript). Although refined analysis for "almost smooth function" is provided in [SZ13], the refined results, even for linear SVM, depend on the assumptions of underlying data distribution (see the discussion in Section 5 and the definition of $N(u)$ in Theorem 16 of [SZ13]).
2. **The technical details of the two methods are different**. [SZ13] and ours do share some similarities at a structual level. However, they differ in the details of the iterative steps. Specifically, [SZ13] directly solves the convex conjugate of the loss function, while our method further decomposes this loss function into simpler components for multi-step updates. Intuitively, our approach further leverages the decomposability and linearity of the loss function to perform multiple iterative updates on the decomposed components.
3. **Improve the practical feasibility of convex conjugate**. In fact, solving and implementing the convex conjugate of a general function is not an easy task, especially when the function is a composition of multiple different functions. In comparison, our proposed ReLU-ReHU decomposition offers better implementation capabilities and properties (e.g., Proposition 1).
4. SDCA cannot handle **constrained** optimization problems.

[SZ13] Shalev-Shwartz and Zhang (2013). Stochastic Dual Coordinate Ascent Methods for Regularized Loss Minimization.

---

### Decision · Program_Chairs · 2023-09-21

**Decision:**

Accept (poster)

**Comment:**

At the end of the discussion phase, all reviewers recommended an acceptance. I would recommend that the authors carefully consider the various comments of the reviewers when preparing the final version, such as carefully discussing the prior work (such as SDCA) and toning down the exaggerated performance claim (there are various standard low-iteration-cost methods that could be applied instead of MOSEK so the 1000x faster claim is not believable).